# Large Language Models as your Personal Data Scientist

## Abstract

Large Language Models (LLMs) have contributed to massive performance improvements for various language understanding and generation tasks; however, their limits are yet to be fully explored for "ill-defined" complex tasks. One such task is conversational data science, where a user can talk to an intelligent agent to explain their data science needs, and the agent will serve the user by engaging in a conversation with them like any human data science would do and, accordingly, formulate and execute precise Machine Learning tasks. Although this is a very ambitious goal, given the recent developments in LLMs, a fully functional conversational data science system seems quite achievable in the near future. Through an in-depth case study in this paper, we delved into the potential of employing LLMs as a solution to conversational data science. We hope that our findings will not only broaden the horizons of NLP research but also bring transformative changes in future AI technology.

## 1 Introduction

The advent of Large Language Models (LLMs) is heralding a new era in the field of Natural Language Processing (NLP) research, opening the door to a plethora of untapped opportunities. These groundbreaking models have showcased their versatility in various tasks, including news summarization, paraphrase comprehension, inference generation, scientific writing improvement, code bug detection, and machine translation Fan et al. (2023). In addition, innovative prompt engineering techniques have significantly extended these capabilities Zhao et al. (2023). Despite these strides, the upper limit of LLMs' potential remains unknown. Testing LLMs in more complex and dynamic scenarios is essential to grasp this boundary, revealing a significant research opportunity.

One such complex and dynamic scenario is conversational data science, where a user can perform data science tasks on their own data sets merely by conversing with an intelligent agent. While such a data science task assistant does not exist yet, given the recent developments in LLMs, a fully functional conversational data science solution seems not only achievable but also quite feasible in the near future. At present, the closest endeavor toward this direction is "Automated Machine Learning" (AutoML), which is mostly inaccessible to the general public. Indeed, as illustrated in Figure 1, the existing AutoML process is fraught with challenges for the end users (e.g., domain experts), who are required to possess a profound understanding of the underlying machine learning tasks, objective functions, optimization techniques, training and validation process, etc. However, this level of understanding often eludes these end users, leading them to rely on the expertise of data scientists to perform these tasks Sarkar et al. (2023). This reliance manifests as time-consuming and inefficient dialogues with data scientists, thus obstructing AutoML from actualizing its potential for true automation Karmaker et al. (2021).

To address these limitations, we envision a novel conversational AI agent called **VIDS** (**V**irtual **I**nteractive **D**ata **S**cientist), capable of assisting users in conducting basic Data Science tasks through intuitive, natural conversations without requiring in-depth knowledge of the underlying core Machine learning (ML) processes. This agent's key functionalities are as follows: 1) accurately comprehend the user's prediction goals and, consequently, 2) formulate a precise ML task, 3) curate data sets and assign model hyper-parameters accordingly, 4) execute an autoML pipeline, and 5) communicate results effectively. To realize VIDS, we utilized

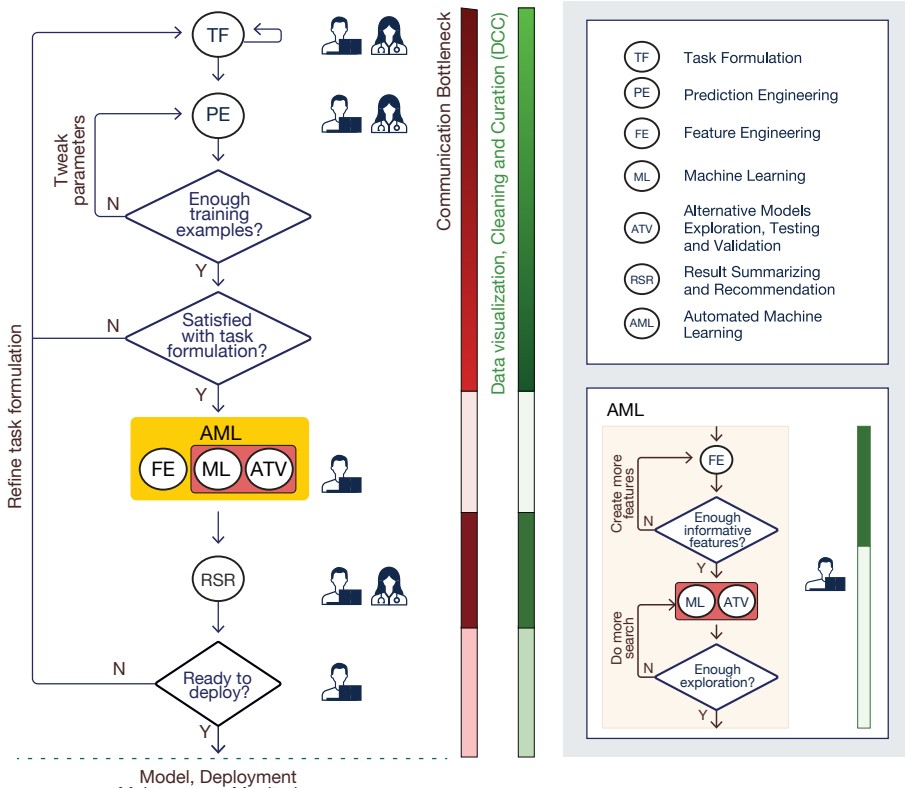

Figure 1: A flowchart showing the end-to-end machine learning process. This figure highlights points of interaction between domain experts and data scientists, along with bottlenecks Karmaker et al. (2021). In this paper, we focus on automating three steps in the chat cycle with the largest communication bottleneck: Task Formulation (TF), Prediction Engineering (PE), and Result Summarization and Recommendation (RSR).

LLMs to build a natural interface between the users and existing AutoML tools (like Scikit-Learn), which in turn, allowed us to approach this ambitious goal with a viable solution.

We designed the architecture of VIDS with four dialogue states: Data Visualization, Task Formulation, Prediction Engineering, and Result Summary and Recommendation. Each state marks a unique conversation phase, impacting the overall user-system interaction. Multiple LLM instances, serving as "micro-agents", ensures a cohesive conversation flow, granting us granular control over the conversation's progression. In summary, we designed and developed an end-to-end system that demonstrates the viability of *Conversational Data Science* by evaluating the potency of LLMs in solving such an ill-defined complex task.

Our main contribution in this paper is to make this connection between two seemingly independent yet complimentary research areas, i.e., LLMs and AutoML, and thus, build a novel solution for realizing Conversational Data Science. In terms of research questions, we explored the following: 1) Is conversational AI a feasible way to perform data science tasks? 2) How accurate are LLMs in framing and solving ill-defined complex data science tasks? 3) What are the common challenges involved with systems like VIDS?

## 2 Related Works

LLMs like ChatGPT [1], BLOOM Workshop et al. (2023), LLama Touvron et al. (2023), PaLM Chowdhery et al. (2022), etc. (Chowdhery et al. (2022); Brown et al. (2020); Zeng et al. (2022); Zhang et al. (2022); Ouyang et al. (2022)), have reshaped the field of NLP research with their broad applicability and impressive performance across diverse tasks. For instance, they have been utilized in automatic news summarization Zhang et al. (2023), efficient healthcare research Sallam (2023), software bug fixing Surameery & Shakor (2023), and machine translation Jiao et al. (2023), showcasing their ability to handle diverse tasks and deliver practical real-world solutions. In benchmarking studies, LLMs have outperformed traditional models like BERT in many tasks and achieved comparable performance in others Zhong et al. (2023).

These models' performances have been further enhanced through various prompting techniques such as chain of thought prompting Wei et al. (2023), self-consistency prompting Wang et al. (2022), synthetic prompting Shao et al. (2023), generated knowledge prompting Liu et al. (2022), and input-output prompting Ma et al. (2023). These techniques improve the models' reasoning, adaptability, and coherence over multiple turns, highlighting their flexible nature and inherent knowledge. Moreover, research like FrugalGPT Chen et al. (2023) has focused on using LLMs more sustainably and efficiently, matching the performance of top LLMs at a significantly reduced cost.

Despite these advancements, the true upper limit of LLMs remains a tantalizing unknown, calling for further exploration of more complex, ill-defined tasks. One such complex task is to provide conversational data science support for end users, which are often ill-defined unseen tasks that could truly test the potential of LLMs. While the machine learning and systems communities have made substantial progress over the past decade in automating various aspects of data science pipelines, some areas still remain underdeveloped. For example, *Data Cleaning and visualization* Ilyas et al. (2015); Chu et al. (2016), *Feature Engineering* Katz et al. (2016); Kanter & Veeramachaneni (2015); Mountantonakis & Tzitzikas (2017); van den Bosch (2017); Khurana et al. (2017); Kaul et al. (2017), *Learning and Parameter Tuning* Bergstra & Bengio (2012); Snoek et al. (2012); Hutter et al. (2011); Bergstra et al. (2011); Bengio (2012); Bergstra et al. (2013); Swersky et al. (2013); Maclaurin et al. (2015), *Alternative Models Exploration, Testing and Validation* Thornton et al. (2013); Feurer et al. (2015); Swearingen et al. (2017); Zoph & Le (2016); Zoph et al. (2017); Liu et al. (2017b;a); Real et al. (2018); Pham et al. (2018); Baker et al. (2017) have all seen significant advancements. Yet, the inherently complex and human-centric parts like Prediction Task Formulation and Result Summarization and Recommendation have not seen the same level of advancement in terms of automation. The limited progress in these areas not only presents a unique research opportunity but also highlights the potential impact of automation in data-reliant industries.

While LLMs have demonstrated their ability to understand intricate language patterns and generate coherent, contextually appropriate responses, they are yet to be fully leveraged for complex human-centric tasks. This underexplored area presents both a challenge and an opportunity. On one hand, it highlights the hurdles such as managing the complexity of handling "ill-defined" tasks, ensuring consistency in responses, and mitigating the tendency of LLMs to generate implausible or 'hallucinated' information Xu et al. (2022); Mündler et al. (2023); Ji et al. (2023). On the other hand, it underscores the untapped potential of LLMs in bridging the automation gap in data science. This work aims to shed light on the potential of this interdisciplinary research area and contribute to the growing body of knowledge on the applications of LLMs in conversational data science.

## 3 Model Architecture

This section delves into the methodology and technical details of VIDS, articulating the intricate interplay between overarching structures and localized nuances. Central to our model are four distinct dialogue states - Data Visualization, Task Formulation, Prediction Engineering, and Result Summary and Recommendation, with each representing a unique phase in the conversation and contributing significantly to the overall user-system interaction. VIDS employs multiple stateless global micro-agents, functioning independently of any state-related data or history, to create an overarching structure that enables fluid transitions throughout the

---

[1]https://openai.com/blog/chatgpt

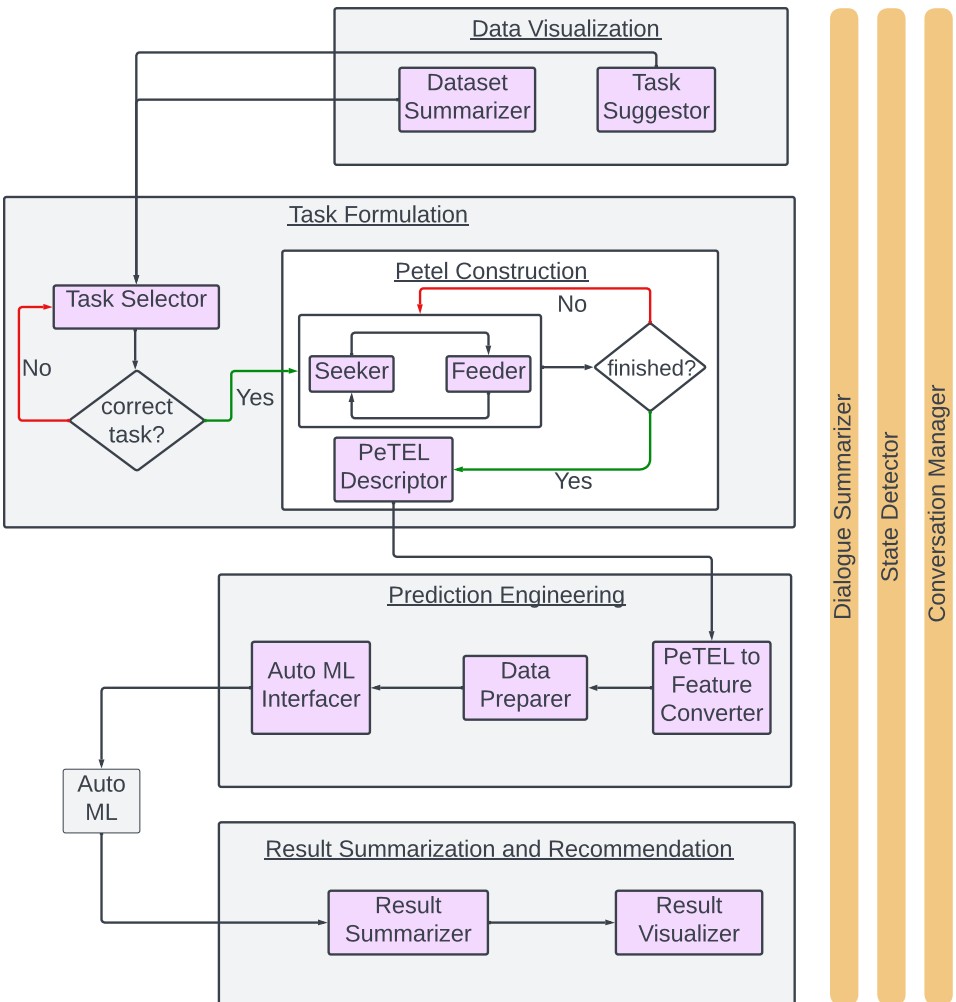

Figure 2: The state diagram of the VIDS dialogue system. The gray boxes are different states of the conversation history, the dark yellow boxes are global micro-agents, and the purple boxes are the local micro-agents.

dialogue, irrespective of the specific state. This stateless design ensures a smooth narrative flow and avoids complications of state-dependent biases or entanglements, thus bolstering the versatility and adaptability of our dialogue system. Alongside these global agents, local micro-agents, each tailored to a specific dialogue state, proficiently handle the nuances of user utterances and conversation contexts, facilitating smooth transitions between states in line with the evolving dialogue. VIDS' strength lies in this symbiotic relationship between the global and local micro-agents across the different dialogue states.

## 3.1 Global Micro-agents

VIDS deploys three global micro-agents across all dialogue states, which are briefly discussed below.

### 3.1.1 State Detector

As a cornerstone of conversation management, VIDS deftly incorporates a number of distinct states, each one reflecting the various phases of interaction with the end user, as illustrated in Figure 2. The system is initialized with the "*Data Visualization*" state, simplifying the data exploration phase. It then progresses to the "*Task Formulation*" state, interpreting the user's intended task. The subsequent "*Prediction Engineering*"

state revolves around crafting training/testing data partitions and initializing model hyper-parameters based on the already defined tasks. The interaction culminates in the "*Result Summarization and Recommendation*" state, summarizing results and offering pertinent recommendations. Leveraging the latest context, current dialogue state, and the user's utterance, the system dynamically infers the next conversation state. It facilitates a seamless dialogue flow, aligning with user needs while enriching their interactive experience. Table 1 (appendix) showcases the unified prompt design guiding LLMs to identify the conversation state and the user intent accurately.

### 3.1.2 Dialogue Summarizer

This global micro-agent distills ongoing conversations into concise summaries, ensuring efficient communication among various micro-agents. It crafts these summaries by integrating the latest user utterance, previous dialogue history, and current responses, maintaining a coherent context throughout the conversation. The unified prompt design for guiding LLMs in summarizing user-VIDS interactions is presented in Table 2 (appendix).

### 3.1.3 Conversation Manager

This micro-agent generates accurate responses by integrating inputs from other micro-agents, maintaining the conversation's context for an effective and seamless dialogue experience for the user. The promoting approach used to steer LLM is demonstrated in Table 3 (appendix).

## 3.2 Local Micro-agents

### 3.2.1 Micro-agents for Data Visualization

First, the user provides a dataset of their choice for customized exploration. Then, the visualization process begins by generating a condensed dataset version, followed by an LLM-driven extraction of insights using the **Dataset Summarizer** micro-agent. Subsequently, the **Task Suggestor** micro-agent, informed by these insights, proposes a suitable Machine Learning task. These cooperative micro-agents ensure efficient dataset exploration and readiness for the Task Formulation phase.

In summary, the Dataset Summarizer and Task Suggestor micro-agents, crucial to this stage, delve into the dataset and propose ML tasks, respectively. These agents, guided by prompting strategies presented in Table 4 and Table 5, pave the way for the Task Formulation stage.

## 3.3 Micro-agents for Task Formulation

Following Data Visualization, VIDS proceeds to the Task Formulation stage. This state is broken down into two interconnected components: *Task Selection* and *PeTEL Representation Construction*, each managed by specialized micro-agents to ensure a precise formulation of the goal ML task.

**Task Selection:** The Task Selection phase, managed by the ***Task Selector*** micro-agent, defines the machine learning task from the pool of suggestions based on the dataset summary and user's data science goals. It provides options like classification, regression, clustering, or user-suggested tasks. This iterative dialogue refines user requirements and aligns tasks with their dataset and goals, as directed by the prompting strategy in Table 6. This micro-agent leverages the conversation summary to properly guide the LLM in selecting a suitable ML model that meets both the dataset's characteristics and the user's needs. This process continues until the user is satisfied and confident with their task choice, enabling personalized, hands-on problem-solving within the micro-agent framework.

**PeTEL Representation Construction:** Upon task selection, VIDS leverages the Prediction Task Expression Language (PeTEL) Karmaker et al. (2021), a concise, slot-value style structured language that encapsulates the core aspects of the chosen machine learning task. The slot-value format of the PeTEL expression provides an unambiguous task description, delineating desired outcomes and key search parameters.

For shaping a more user-tailored experience, VIDS uses the PeTEL Representation Construction group—a team of micro-agents working in concert, emphasizing precision and user satisfaction. Key components among them are the ***Seeker*** and ***Feeder*** micro-agents, which facilitate an iterative dialogue with the user to populate PeTEL slots. The ***Seeker*** drives the conversation toward Task formulation, ensuring no aspect of the task is missed (refer Table 7 in appendix). In parallel, the ***Feeder*** assimilates user responses into PeTEL slots, interpreting user inputs for precise task specification (prompts detailed in Table 8 of the appendix).

Collectively, the ***Seeker*** and ***Feeder*** steer a dynamic dialogue until PeTEL accurately mirrors the user's intent. Finally, the PeTEL Descriptor micro-agent articulates the populated PeTEL in layman's terms, reinforcing user understanding (see Table 9) and active participation. Thus, the PeTEL Construction process yields an effective, user-specific task representation, paving the way for subsequent machine learning stages. Refer to Listing 1 in the appendix for a filled PeTEL example.

### 3.4 Micro-agents for Prediction Engineering

The Prediction Engineering phase is the bridge between abstract problem representation and a practical prediction model, composed of three steps: converting PeTEL to features, executing data preparation, and interfacing with AutoML systems.

First, the ***PeTEL to Feature Converter*** micro-agent transforms the task representation into tangible features, readying the problem description for computational processing. Next, the ***Data Preparer*** micro-agent refines the dataset, handling missing data, outliers, and categorical variable encoding to facilitate the application of machine learning algorithms. Finally, the ***AutoML Interfacer*** micro-agent uses the prepared training and testing data sets with AutoML systems. Utilizing these platforms' automation, it selects, optimizes, and trains an appropriate machine-learning algorithm.

### 3.5 Micro-agents for Result Summary

Data scientists traditionally present findings and advise domain experts on strategies, but this stage is mostly manual. Therefore, automation of Result Summarization and Recommendation is indeed very challenging. Nevertheless, we propose a hypothetical micro-agent ***Result Summarizer*** that will generate a comprehensive summary of findings by employing AutoML libraries like Auto-SKLearn to train multiple models. This summary will enable users to compare and identify the most effective solution, rapidly understanding the core findings. Future versions of VIDS will implement such a ***Result Summarizer*** micro-agent to streamline the Result Summary and Recommendation phase.

Additionally, As part of its future vision, VIDS aims to introduce the ***Result Visualizer*** micro-agent, designed to elevate users' comprehension through visual representations of outcomes. This advanced mechanism will generate suitable visualizations, such as performance metrics and feature importance, providing users with an intuitive and enhanced understanding of the findings. Furthermore, our shoot-the-moon objective is to optimize decision-making through interactive dialogue, where the system recommends the optimal model based on the ongoing conversation. This personalized approach is set to simplify the process, empowering users to make well-informed decisions.

## 4 Case study and Qualitative Evaluation

This section reports a qualitative evaluation of LLMs in performing complex tasks through an in-depth case study. We exclusively focus on ChatGPT for this case study and utilize the Student Performance dataset[2] for our analysis. Our evaluation revolves around three main criteria. Firstly, we examine the *usability* and *efficacy* of LLMs in facilitating seamless conversations for data science tasks by analyzing the chat cycle between users and the VIDS system. Secondly, we examine the interaction among the micro-agents themselves, each equipped with its unique LLM instance. This analysis reveals the intricacies of micro-agent dynamics in performing complex "ill-defined" tasks. Lastly, we assess the versatility and resilience of the TeLER Prompt Taxonomy Santu & Feng (2023), which we followed to design prompts that were fed

---

[2]https://www.kaggle.com/datasets/larsen0966/student-performance-data-set

to the LLMs serving as various micro-agents, demonstrating the practicality and applicability of LLMs in demanding scenarios. Through this in-depth qualitative examination, we aim to provide a panoramic view of the essential role and effectiveness of LLMs in navigating complex tasks.

## 4.1   Usability and Efficacy of LLMs

To evaluate the usability and efficacy of LLMs as a solution to conversational data science, we focused on the essential dynamics of the user-system chat cycle, which is pivotal for effective communication within VIDS. A deep dive into this cyclical process has elucidated how LLMs orchestrate fluid, efficient communication. This scrutiny has further shined a light on the user experience, underscoring the prowess and utility of the LLMs in conversational data science. Table 10 (in the appendix) presents a tangible snapshot of user-system interactions, exemplifying the successful execution of the user's goal task via natural conversation.

## 4.2   Micro-Agents' Roles and Interactions

In this subsection, we explore the interplay among the micro-agents within the four primary states of the system. Each micro-agent, armed with an LLM in the background, contributes to the system's overall execution of complex tasks. Through an analysis of these internal dynamics, we sought to deepen our comprehension of LLM functionality.

Data visualization, the initial state, presents data in a visually intuitive format, unveiling inherent complexities and patterns. The effectiveness of this representation is hinged on the interaction of *Data Summarizer* and *Task Suggestor* micro-agents, as illustrated in Table 11.

Next, during the Task Formulation state, the task definition and specifics are established. The interplay among the *Task Selector*, *Seeker*, *Feeder*, and *PeTel Descriptor* micro-agents was pivotal in shaping the task, significantly influencing the eventual success of execution. The specifics of these interactions are presented in Table 12.

Moving forward, in the Prediction Engineering state, the dataset is tailored according to the formulated task. The collaboration among micro-agents (*PeTEL-to-Attribute-Converter*, *Data Preparer* and *AutoML Interfacer*) in this phase directly impacts the dataset preparation, influencing the accuracy of predictions. Further details are provided in Table 13 (appendix), fostering a comprehensive understanding of micro-agents' roles within this critical AutoML phase.

Finally, VIDS interfaces with AutoML tools like AutoSKLearn to train selected models. From these training performances, VIDS generates result summaries and recommendations aligned with user preferences, as depicted in Table 14. Future work involves customizing such recommendations based on the user's business goals.

## 4.3   Versatility and Resilience of TeLER

Understanding the interaction between humans and AI at different levels of detail, as defined by Santu & Feng (2023) as the TeLER taxonomy, is vital for effective collaboration, especially in complex tasks. Below, we discuss the role of different levels of prompt details (as defined by the TeLER taxonomy) for three micro-agents— State Detector, Dialogue Summarizer, and Conversation Manager—which work together to facilitate efficient user-AI dialogue.

**State Detector micro-agent:** Table 15 displays LLM's responses to each TeLER level (0-5). On analyzing this table, it becomes clear that as task specificity in the prompt increases, the results become more targeted. Low-detail prompts (Levels 0 and 1), due to their lack of precision, fail to generate the desired outputs, emphasizing the effectiveness of detailed prompt articulations.

**Dialogue Summarizer micro-agent:** Table 16 displays the LLM's responses to each TeLER level. A detailed review of this table offers several key findings. Firstly, LLM appears to bypass embedded instructions or subtasks when provided with high-complexity prompts. For instance, while a level 4 prompt necessitates an explanation of the response from LLM, a level 5 prompt also demands the provision of evaluation criteria.

Nevertheless, as evidenced in Table 16, LLM often neglects these subtasks, indicating a potential limitation with processing long prompts. Secondly, despite the micro agent's primary goal of dialogue summarization and the expectation to emphasize the most recent utterance, LLM often generates a more generic summary, consistently incorporating dataset-related information throughout the dialogue. This behavior suggests a bias towards longer dialogue turns.

**Conversation Manager micro-agent:** As presented in Table 17, LLM responses tend to become verbose with highly detailed prompts, implying an optimal length for prompts that can avoid overwhelming users in terms of cognitive load.

## 5 Discussion

This section discusses the strengths and shortcomings of LLMs for realizing conversational data science based on the observations from our case study.

### 5.1 The Strengths

In terms of strengths, our case study revealed that LLMs can- 1) achieve more automation within the end-to-end ML pipeline, 2) enhance the precision of AI responses, 3) increase the efficiency in communication with users, 4) provide a more controlled dialogue flow, and 5) contribute to the democratization of Data Science technology.

**LLMs in Automation:** In terms of automating an end-to-end AutoML pipeline, especially during the *Task Formulation* and *Prediction Engineering* stages, the utility of LLMs cannot be overstated. With their robust capabilities, LLMs can significantly streamline these highly interactive complex processes (see Tables 1-17), leading to broader accessibility of Data Science to the general public.

**Precision Control:** We found that decomposing larger tasks into manageable, specific micro-agents helped to ensure precise LLM responses. Our findings also suggest that using multiple targeted prompts, as opposed to a single comprehensive one, leads to better control over the LLMs' responses, making them more focused and reducing the likelihood of errors or misunderstandings.

**Efficiency:** The precision garnered from micro-agent decomposition also increases the efficiency of LLMs. As an example, the *Seeker* micro-agent, as highlighted in the 7th row of Table 12, demonstrates high efficiency in understanding the current dialogue state and crafting appropriate questions. We believe such efficiency emerged from each connected micro-agent (e.g., *Feeder*, *PeTEL Descriptor* etc.) performing their assigned tasks precisely and coordinating with each other properly.

**Dialogue Flow Control:** LLM (ChatGPT) performed commendably in terms of overall dialogue flow control by carefully balancing multiple factors, such as the utterance, context, responses from various micro-agents, user intent, and the current state of the conversation. The dialogue flow was best controlled when Level 3 prompts (defined by the TeLER taxonomy) were used, as shown in Table 17 (appendix). Indeed, LLM skillfully extracted relevant information and steered the conversation in the right direction, which is very promising.

**LLM and Democratization of Data Science:** Based on our case study, we believe a product level realization of *VIDS* will enable the general public/domain experts to directly engage with the fascinating world of predictive analytics without worrying about the underlying detailed knowledge of machine learning. *VIDS* will also automate many of the routine tasks performed by a *human* Data Scientist. Although this is a highly ambitious goal, if reached, it will expedite the implementation of ML systems allowing Data Science to appeal to a far broader audience across various domains.

### 5.2 The Shortcomings

In this section, we scrutinize and discuss the challenges that LLMs face with "ill-defined" complex tasks and prompts, viewing these complexities as a means to test their upper limits.

**Shortcomings in State Detection:** The State Detector micro-agent demonstrated lapses, struggling to interpret complex prompts, despite distinct delineations between states (Table 1). For instance, in one case, the model erroneously identified "Model Training" as the current state instead of the intended "Task Selection" state where the user utterance clearly conveyed the goal of performing a classification task (Table 11). To enhance accuracy, we introduced a refined prompt design listing potential subsequent states, resulting in an improvement of the state selector's precision.

**Shortcomings in Dialogue Summarization:** The *Dialogue Summarizer* micro-agent brought a set of hurdles, especially by "hallucinating" irrelevant content. To address this issue, we provided few-shot examples of conversations and summaries, aiming for LLM to adopt that summarizing pattern. However, during evaluation, LLM still failed in some cases, erroneously weaving the examples into the summary rather than focusing on the latest input.

**Shortcomings in Handling Complex Prompts:** Our investigation into LLM's responses to varying prompt complexities, as illustrated in Table 17, revealed noteworthy patterns. Firstly, LLM tends to generate verbose responses when faced with detailed prompts, potentially inundating users with excessive information. Secondly, with complex prompts to the *Conversation Summarizer* micro-agent, LLM often overlooks embedded instructions or subtasks. For instance, a level 4 prompt asks for a self-explanatory LLM response, while a level 5 prompt additionally requires following some evaluation criteria. However, LLM frequently sidesteps these subtasks, highlighting a potential challenge with handling intricate prompts (see Table 16).

### 5.2.1 Summary and Future Directions

Our exploration into LLMs' role in Conversational Data Science reveals their promise and the challenges surfacing at their operational boundaries. While LLMs showcase the potential to enhance precision, efficiency, and dialogue flow management in automated systems, their full capacities are yet to be unearthed. However, we also encountered difficulties with LLMs in state detection, dialogue summarization, and intricate prompt handling. For example, the *Dialogue Summarizer* micro-agent, tasked with summarization, tends to generate generic summaries with a potential bias towards lengthy dialogues over the most recent utterances. On top of that, overly complex prompting techniques generated responses that were voluminous, turned out to be counterproductive. These findings emphasize the need for further exploration and refinement to optimize the balance between task complexity and LLM performance and to reveal the true capability of these models in automation. As we strive to realize the full potential of LLMs, we remain committed to pushing their boundaries, thereby aiding in the advancement and democratization of Data Science.

## 6 Conclusion

This paper illuminates our journey into the uncharted potential of Large Language Models (LLMs) toward the ambitious goal of conversational data science, an arena characterized by complex, ill-defined tasks. We have successfully architected an innovative concept of personal data scientist, VIDS, capitalizing on the capabilities of LLMs to use natural language as the primary interface. VIDS' architecture encompasses four distinct dialogue states, each symbolizing a unique phase of the conversation and crucially impacting the user-system interaction. Further, the concept of global micro-agents, forming a holistic structure and ensuring a coherent dialogue throughout the session, was introduced. Alongside these, we've integrated state-specific local micro-agents, instrumental in delivering VIDS' functionality.

Although VIDS exhibits advanced capabilities, it's also very important to acknowledge areas of challenge. Even as we've applied amendments to address these issues, experiencing certain improvements, these obstacles highlight the need for further exploration and development. Identifying and rectifying these failures is key to enhancing the model's robustness, efficiency, and overall performance, thereby bringing us closer to harnessing the full potential of LLMs.

Our study serves as a crucial stepping-stone towards our ultimate vision of a conversational data science assistant that can effectively understand and engage with its users. The knowledge garnered from this research will guide our future efforts to optimize LLMs. We are steadfast in our belief that the continued

refinement of these models will catalyze more intuitive and effective human-AI interactions, revolutionizing how we approach complex tasks and data analysis and unveiling the true potential of LLMs for automation.

## 7    Limitations

While our explorations into the landscape of Large Language Models (LLMs) and conversational data science have been pioneering, we recognize our work is in its early stages and is not without its limitations. We've highlighted several areas for improvement and potential paths forward, each presenting a unique challenge and opportunity to propel VIDS and the field of LLMs into a new dimension.

**Single Model Dependence:** Presently, our investigation leans heavily on ChatGPT, bypassing a comparative examination of various LLMs. A promising avenue for future inquiry involves contrasting performance of multiple LLMs tackling such intricate tasks.

**Automated Result Summarization & Model Recommendation:** Our system lacks automated generation of Result Summaries and Model Recommendations, both potential game-changers. Future enhancements could include dynamic visual adjustments responsive to conversation context and model suggestions tailored to individual business objectives and requirements.

**Scalability & Performance:** With the ongoing expansion of VIDS' features, ensuring robust scalability and prompt responsiveness emerges as a critical concern. Future efforts should focus on refining system optimization to balance between large-scale deployments and performance.

**VIDS Personalization:** At present, VIDS misses personalization, a feature that can cater to individual user preferences. Future versions should incorporate adaptable user-specific customization for a more engaging experience.

**Multilingual Adaptation:** VIDS' current English-only operation restricts its global outreach. Future updates should incorporate multilingual capabilities, broadening the appeal and accessibility of VIDS.

**Interpretability & Transparency:** VIDS currently doesn't elucidate its underlying reasoning, possibly causing user apprehension. Future iterations should emphasize improved model interpretability and transparency to build user confidence.

**Ethical Considerations & Bias Detection:** VIDS lacks mechanisms to detect or counterbalance biases. Future research should dedicate efforts to uncover potential biases and devise strategies for equitable, ethical interactions.

In the complex landscape of conversational data science, navigating these limitations offers unique opportunities to pressure test and expand the capabilities of Large Language Models (LLMs). These challenges, intrinsic to ill-defined tasks, aren't merely obstacles but pivotal springboards that propel our understanding of LLMs. Tackling them not only refines our approach towards task automation with VIDS, but also contributes substantially to the broader field of NLP. This continuous pursuit, aimed at enhancing AI-human interactions in complex and ill-defined tasks, fuels innovative strides in NLP research, ensuring we remain at the forefront of this rapidly evolving domain.

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

# A  Global Micro-agents

In this section we discuss about the global micro-agents and how our prompt design has helped to actualize the those agents.

## A.1  State Detector:

As shown in Figure 2, VIDS integrates a variety of well-defined states, each corresponding to the different stages of a conversation. The initial state is "data visualization", which centers around the presentation of data in a comprehensible and approachable manner. This transitions into the "task formulation" state, wherein the focus shifts to defining and structuring the task or problem the user wishes to address. Following this, the system moves into the "prediction engineering" state. In this phase, the system focuses on constructing and implementing predictive models that are based on the tasks as defined in the previous stage. Finally, the conversation arrives at the "result summarization and recommendation" state. Here, the system offers a succinct summary of the results, coupled with relevant recommendations based on the outcomes.

Our system employs a micro-agent designed to dynamically determine the user's intent by considering not only the immediate context and user's utterance, but also the current dialogue state. This micro-agent, taking as input the current context, conversation state, and user input, decides whether the user wishes to proceed to the next state of the dialogue. By accurately aligning with the user's needs and objectives, this approach ensures a smooth flow of conversation while also providing an engaging user experience. The design of our system, therefore, strikes a balance between addressing user needs and enriching their interaction with the system. The unified prompt design used to guide the LLM in correctly identifying the current state of the conversation and the user's intent is presented in Table 1.

### A.2 Dialogue Summarizer:

Our system incorporates a micro-agent responsible for generating concise summaries of the ongoing conversation. This crucial feature enhances effective communication among various micro-agents. Given the dialogue history as input, this micro-agent takes into account the latest user utterance, preceding conversation history, and the current response from another micro-agent. Consequently, it generates a new dialogue summary that preserves coherence and context throughout the conversation. The unified prompt design used to guide the LLM in summarizing the interactions between the user and VIDS is displayed in Table 2.

### A.3 Conversation Manager:

Our system includes a conversation management micro-agent that weaves together the outputs from relevant micro-agents, generating a coherent, overarching dialogue. It accepts a variety of inputs including the context, the conversation state, the utterance, current intent, and the response from the micro-agent. By effectively maintaining the dialogue's structure and context, it assures a seamless user experience and adept task execution. The unified prompt design that steers the LLM in this process is shown in Table 3.

## B  Local Micro-Agents

In this section we discuss about the micro-agents that works together to make sure there is seamless conversation flow at each stage.

### B.1 Data Visualization

Upon dataset selection, VIDS efficiently generates a summarized version via LLM and its Dataset Summarizer and Task Suggestor micro-agents, providing a thorough understanding of the dataset and suggesting potential Machine Learning tasks.

#### B.1.1 Dataset Summarizer micro-agent:

The Dataset Summarizer micro-agent, at the core of the Data Visualization stage, employs a purposefully designed prompt to extract insights from the provided dataset. This agent accepts the dataset as its input and delves into its structure and content, gleaning an array of insights that furnish users with a comprehensive understanding of the dataset's potential applications. This extraction process is guided by the unified prompt design, as shown in Table 4, ensuring a thorough, user-friendly data analysis experience.

#### B.1.2 Task Suggestor micro-agent:

The Task Suggestor micro-agent, taking as its input the dataset summary, dovetails with the Dataset Summarizer to propose pertinent Machine Learning tasks. Leveraging a unified prompt design, detailed in Table 5, it guides the LLM to yield effective task suggestions. This capability, integral to the Data Visualization stage, enriches user interaction and effectively paves the way for the subsequent Task Formulation stage.

### B.2 Task Formulation

Following the Data Visualization stage, VIDS proceeds to Task Formulation. This section is broken down into two interconnected components: Task Selection and PeTEL Construction. , each managed by specialized micro-agents to ensure a thorough and user-oriented formulation of the machine learning task. Task Selection is managed by the Task Selector micro agent where PeTEL Construction component is handled with a group of micro-agents that consists of the three micro-agents: Feeder, Seeker, and PeTEL Descriptor. A sample populated PeTEL, demonstrating the iterative process of filling out the different components, is available in Listing 1.

### B.2.1 Task Selector micro-agent:

Functioning within our system, the task selection micro-agent navigates through user conversations to pinpoint an apt machine learning problem from a variety of available options. It also provides valuable assistance in choosing a model best-suited to user needs. The micro-agent operates using inputs such as the context of the dialogue and user utterance, along with understanding user requirements and factoring in the dataset's characteristics. This enables it to select from diverse model types, including but not limited to, 'classification', 'regression', 'clustering', 'dimensionality reduction', and 'anomaly detection'. By ensuring alignment between the chosen problem, model, and user's objectives, it offers personalized recommendations, thereby enhancing user engagement. The unified prompt design guiding the LLM during this process is presented in Table 6.

### B.2.2 Seeker micro-agent:

As an integral member of the PeTEL Construction micro-agent group, the Seeker micro-agent conducts user interactions to fill the next slot in the PeTEL representation. It takes two main inputs: the context of the dialogue and the current PeTEL. Through its systematic guidance, it ensures each unfilled slot is populated, facilitating a comprehensive and accurate task formulation. Table 7 demonstrates the unified prompt design utilized to direct the LLM effectively in querying about specific unfilled slots in the PeTEL expression.

### B.2.3 Feeder micro-agent:

Our PeTEL Construction micro-agent group includes the Feeder micro-agent, a crucial component that populates the PeTEL representation with user responses. This micro-agent accepts inputs such as the context of the dialogue, the user utterance, and the current PeTEL, ensuring accurate interpretation and integration of user's utterances into problem formulation. Table 8 presents the unified prompt design that guides the LLM to effectively fill unfilled slots in PeTEL expressions from user interactions.

### B.2.4 PeTEL Descriptor micro-agent:

Rounding out the PeTEL Construction micro-agent group, the PeTEL Descriptor micro-agent communicates the fully populated PeTEL expression back to the user in natural language. It takes as input the fully populated PeTEL, and its function underlines VIDS' user-centric approach by ensuring that the user completely comprehends the formulated task. The unified prompt design that guides the LLM in this process is depicted in Table 9.

```
{
  problem_type: classification,
  target_variable: delay_severity,
  features: [departure_airport, arrival_airport, airline, scheduled_departure_time
      , scheduled_arrival_time, weather_conditions],
  dataset_size: 10000/Default,
  performance_metrics: [accuracy, precision, recall, f1_score, confusion_matrix],
  validation_method: cross_validation,
  classification_methods: [logistic_regression, decision_tree_classifier,
      random_forest_classifier, svm_classifier, knn_classifier, xgboost_classifier,
       naive_bayes],
  data_filters: [
    {column: delay_duration, condition: greater_than, value: 15},
    {column: departure_airport, condition: equals, value: JFK}
  ],
  business_goals: [reduce customer complaints, optimize scheduling, improve
      airport operations],
  additional_requirements: [robust to outliers, handle class imbalance],
  model_preferences: interpretable
}
```

Listing 1: Sample populated PeTEL for classification task based on FlightDelay dataset (one of our demo datasets).

## B.3   Prediction Engineering

Following Task Formulation, the journey progresses to Prediction Engineering, a fundamental stage where the system transforms the problem representation into a tangible prediction model. This phase is composed of three micro-agents: PeTEL to Feature, Data Prepper, and AutoML interfacer. Each agent is crucial in bridging the gap between the problem's conceptual representation and its practical implementation.

### B.3.1   PeTEL to Attribute Converter Micro-Agent:

The PeTEL to Feature conversion is the first step in the Prediction Engineering process. Here, the PeTEL representation, which succinctly describes the machine learning task, is translated into features that can be used by the prediction model. This process ensures that the machine learning algorithms can interpret and work with the problem definition, turning the abstract task representation into concrete, computable features.

### B.3.2   Data Prepper Micro-Agent:

Once the features are defined, the next step is Data Cleaning and Preparation. This stage involves pre-processing the dataset to ensure it's suitable for the prediction model. Common procedures during this phase include handling missing data, dealing with outliers, and encoding categorical variables. The goal is to produce a clean, well-structured dataset that can be readily consumed by downstream machine learning algorithms, maximizing the potential for accurate and meaningful predictions.

### B.3.3   AutoML interfacer Micro-Agent:

The final step in the Prediction Engineering phase is interfacing with AutoML systems. AutoML platforms automate the process of applying machine learning to real-world problems, making the technology accessible to non-experts and improving efficiency of experts. In this step, the prepared dataset is fed into an AutoML system, which automatically selects the most suitable machine learning algorithm, optimizes its parameters, and trains the model. The result is a robust prediction model that is ready to generate insights from new data, bringing the conceptual machine learning task to fruition.

## B.4   Result Summary and Recommendation

We aim to automate and enhance the final phase of VIDS, the Result Summary and Recommendation, with systematic structuring and incorporation of Result Summarization and Visualization to facilitate informed decision-making in data science.

### B.4.1   Result Summarizer Micro-Agent:

Currently, we have implemented the Result Summarization micro-agent, where the system produces a comprehensive summary of the findings once the machine learning tasks have been executed. Utilizing an AutoML library such as Auto-SKLearn, the system trains all specified models, equipping users with a broad comparison to discern the most effective solution. This process distills the results into an accessible format, enabling users to grasp the essence of the findings quickly.

### B.4.2   Result Visualizer Micro-Agent (Future work):

Looking forward, we aim to implement the Result Visualization micro-agent. Visualizing the outcomes can significantly aid users' understanding and facilitate more informed decision-making. We plan to develop a

process that generates suitable visualizations based on the results, such as performance metrics or feature importance, offering a more intuitive perspective of the findings.

Additionally, we aspire to further optimize the decision-making process, by having the system engage in an interactive dialogue with the user, recommending the most suitable model based on the conversation. This personalized approach would empower users to make informed decisions, streamline the process, and enhance the effectiveness of their machine learning endeavors.

| Prompt Design | |
|---|---|
| **System setup** | |
| The AI assistant has been designed to understand and categorize user input by detecting the user's intent and conversation state. When processing user input, the assistant must identify the intent from one of the following pre-defined options: 'Get dataset info', 'Get dataset trend', 'Select problem', 'Formulate problem', 'Problem execution', or 'chitchat'. It is essential to note that multiple instances of the same intent type are not permitted. If the user input is unclear or cannot be parsed accurately, the assistant should classify it under the 'chitchat' intent, indicating that the input requires further clarification or does not align with the available intent categories. The intent also depends on the current state of the conversation. The rules for state transition are as follows:
current state: data visualization
next available states: data visualization, task selection
current state: ask selection
next available states: ask selection, task formulation
current state: task formulation
next available state: task formulation, model training.
current state: model training
next available state: model training.
The agent MUST response with the following json format: {"intent": " ", "current_state": " ", "next_state": " " } | |
| **Demonstration** | |
| User | Taking into account the given context { In this dialogue, the AI assistant provided information on suitable machine learning tasks for three different datasets: airlines, flights, and airports. For the flights dataset, the assistant suggested that classification and regression would be suitable tasks. Classification could be used to predict flight delays or cancellations, while regression could be used to predict the amount of delay. The user expressed interest to know more about the dataset.}, the conversation state { dataset_understanding } the utterance { What details are included in the flight delay dataset? }, identify my current intent and next state of conversation. Please remember to only response in following format predefined json format without any additional information. Carefully examine the utterance and think about how the context might influence the current utterance, leading you to determine my present intent and next state. |
| LLM | {"intent": "Get dataset info", "current_state": "dataset_understanding", "next_state": "dataset_understanding"} |
| User | Taking into account the given context { In this dialogue, the AI assistant provided information on suitable machine learning tasks for three different datasets: airlines, flights, and airports. For the flights dataset, the assistant suggested that classification and regression would be suitable tasks. Classification could be used to predict flight delays or cancellations, while regression could be used to predict the amount of delay. The user expressed interest in the flights dataset and asked if it could be formulated as a time series problem, but the assistant did not provide a response to this question. }, the conversation state { dataset_understanding } the utterance { I want to predict if a flight will be delayed or not }, identify my current intent and next state of conversation. Please remember to only response in following format predefined json format without any additional information. Carefully examine the utterance and think about how the context might influence the current utterance, leading you to determine my present intent and next state. |
| LLM | {"intent": "Select problem", "current_state": "dataset_understanding", "next_state": "problem_selection"} |
| **Directive** | |
| Taking into account the given context {context}, the conversation state {conversation state} the utterance {user input}, identify my current intent and next state of conversation. Please remember to only response in following format predefined json format without any additional information. Carefully examine the utterance and think about how the context might influence the current utterance, leading you to determine my present intent and next state. | |

Table 1: The details of prompt design for the State Detector micro-agent. In the directive, the {context}, {conversation state}, and {user input} are placeholders which will be replaced dynamically in different stage of conversation

| Prompt Design |
|---|
| **System setup** |
| Given the dialogue between user and assistant, the AI assistant summarizes the dialogue summary. The AI agent should not leave out any crucial information. The goal of this summary generation is not being precise, rather the goal should be to contain all crucial information. if the previous dialogue is empty then you should return the current user utterance. |
| **Directive** |
| Given the context as {context}, latest user utterance as {utterance} and previous response as {response}, summarize the following dialogue. You should not exclude any important information. {history} |

Table 2: The details of prompt design for the Dialogue Summarizer micro-agent. In the directive, {history}, {context}, {utterance} and {response} are placeholders which will be replaced dynamically during the conversation

| Prompt Design |
|---|
| **System setup** |
| The AI assistant serves as a virtual data scientist, designed to engage with users and comprehend their objectives. The purpose of this interaction is to develop a machine learning task tailored to the user's data. To achieve this, the assistant will collaborate with various micro agents, each performing specialized tasks to support the primary agent. The assistant will receive context, utterances, dataset summaries, and micro agent responses as input, and should aim to steer the conversation towards the goal. The following micro agents will aid the assistant, providing their output as input to the AI agent for further processing and integration. Depending on the current conversation state, different micro agents will be activated to provide their respective responses: |
| Intent Detector: Identifies the user's intent from a list including 'Get dataset info', 'Get dataset trend', 'Select problem', 'Formulate problem', 'Problem execution', and 'Chitchat'. The detected intent will be used to determine the direction of the conversation. |
| State Selector: Determines the conversation state, choosing from "data_visualization", "task_selection", "task_formulation", or "task_execution". The chosen state helps the AI agent to adapt its responses and maintain a coherent discussion flow. |
| Task Selector: Selects an appropriate ML task from options such as "classification", "regression", "clustering", "dimensionality reduction", "anomaly detection", and "time series". The selected task guides the AI agent in suggesting relevant solutions to the user. |
| Task Formulator: Constructs the ML task by utilizing a slot-value filling process. The formulated problem, complete with specified parameters, is then provided to the AI agent, which can assist the user in refining or executing the task. |
| **Directive** |
| Taking into account the given context [context], the conversation state {state} the utterance {input}, current intent {intent} and the response from the {micro-agent} micro-agent {MA_resp}, provide appropriate response to the user to carry the conversation to its goal which is formulating a ML task based on user demands. |

Table 3: The details of prompt design for the Conversation Manager micro-agent. In the directive, {state}, {input}, {micro-agent}, and {MA_resp} are placeholders which will be replaced dynamically during the conversation.

| Prompt Design |
|---|
| **System setup** |

**Table 4 – continued from previous page**

| Prompt Design |
| --- |
| You are an AI agent who will provide a conprihensive summary of a given dataset. Your task is to provide a comprehensive summary of a given dataset in a strict "JSON" format. |

The summary MUST include the following informations:

1. dataset summary: the summary of the given dataset in natural language

2. column: it will list all columns and give a brief description about that column

3. Row: AI agent will select a row at random and describe what the row means in natural language

4. Trend: In natural language the AI agent will write the trends that can be found from the given dataset.

The response should be in a strict JSON format as follows: {"summary": "...", "columns": ["name": "col1", "description": "...", "name": "col2", "description": "..."], "row": "description of a random row", "trend", "..."}

Please make sure to provide clear and concise descriptions in natural language to facilitate understanding for non-technical users.

**Directive**

Please provide a comprehensive summary of the given dataset. The response MUST be in JSON format NOTHING ELSE. Use the following dataset: {dataset}.

Table 4: The details of prompt design for the Dataset Summarizer micro-agent. In the directive, the {dataset} is a placeholders which will be replaced a miniature version of the user provided dataset.

| Prompt Design |
| --- |
| **System setup** |

The AI agent must analyze the provided dataset summary and recommend appropriate machine learning (ML) tasks. Based on the summary, column descriptions, row information, and any observed trends, the agent should suggest at least two suitable ML task from the following task list: ["classification", "regression", "clustering", "dimensionality reduction", "anomaly detection", "time series"]. For each ML task the agent chooses a clear rationale must be provided which may include an explanation of why the chosen task aligns with the dataset, and a concrete example of how the task can be formulated.

**Directive**

Suggest ML tasks based on the following dataset summary: {summary}

Table 5: The details of prompt design for the Task Suggestor micro-agent. In the directive, the {summary} is a placeholders which will be replaced by the dataset summary of the user provided dataset.

| Prompt Design | |
|---|---|
| **System setup** | |
| The AI assistant is designed to comprehend the user's needs through conversation and assist them in selecting a suitable machine learning model for formulating a Machine Learning problem. The assistant must choose the appropriate model from the provided list: ["classification", "regression", "clustering", "dimensionality reduction", "anomaly detection"]. The assistant should consider the user's problem, requirements, and dataset, which may be found in the dialogue context, to recommend the best model tailored to their specific needs. | |
| **Demonstration** | |
| **User** | I want to predict whether a flight will be delayed or not based on factors like weather conditions and previous delays. |
| **LLM** | {'model': 'classification', 'reason': 'A classification model can be used to categorize flights as delayed or not delayed based on the input features, such as weather conditions and previous delays.'} |
| **User** | I need to find groups of flights with similar delay patterns, considering variables like departure airport, airline, and time of day. |
| **LLM** | {'model': 'clustering', 'reason': 'A clustering model can help identify groups of flights with similar delay patterns by analyzing variables like departure airport, airline, and time of day, without requiring labeled data.'} |
| **Directive** | |
| Please choose a most suitable model the conversation context: [context] and my latest utterance {input}. The output must be in a strict JSON format: {"model": "model name", "reason": "your detail reasons for the choice"}. | |

Table 6: The details of prompt design for the Task Selector micro-agent.
In the directive, the [context], and {input} are placeholders which will
be replaced dynamically during the conversation

| Prompt Design | |
|---|---|
| **System setup** | |
| The AI assistant functions as a virtual data scientist, aiming to fill in any missing values in a given JSON object by asking the user about them. In order to not overwhelm the user, the assistant should focus on one unidentified slot at a time. With the JSON object, dataset summary, and the next empty slot provided, the agent should formulate a specific and clear question based on this information to accurately identify the missing value. The dataset summary is as follows: {dataset_summary} | |
| **Demonstration** | |

**Table 7 – continued from previous page**

| | Prompt Design |
|---|---|
| **User** | In order to obtain information about the next unidentified slot in the given JSON object |
| | { |
| | problem_type: time_series_forecasting, |
| | target_variable: null, |
| | forecast_horizon: null, |
| | business_goals: null, |
| | granularity: null, |
| | features: null, |
| | time_range: null, |
| | model_preferences: null, |
| | performance_metrics: null, |
| | validation_method: null, |
| | additional_requirements: null, |
| | time_series_methods: null, |
| | data_filters: [ |
| | column: null, condition: null, value: null, |
| | column: null, condition: null, value: null |
| | ] |
| | } |
| | The agent should ask the user a clear and specific question. Considering the provided context (User wants to formulate a time series forecasting problem by using the flight delay dataset. He is interacting with the agent and the agent is now asking questions to the user to formulate the problem), please formulate a suitable query to gather the necessary data for the missing slot. |
| **LLM** | based on the provided context and the current state of the JSON object, the next unidentified slot is the **target_variables** slot. What specific variable are you trying to forecast using the flight delay dataset? This is also known as the target variable. For example, are you trying to forecast the number of flights that will be delayed, or the total duration of delays in minutes? |

| **Directive** |
|---|
| In order to obtain information about the next unidentified slot in the given JSON object ({petel}), the agent should ask the user a clear and specific question. Considering the provided context ({context}), please formulate a suitable query to gather the necessary data for the missing slot. |

Table 7: The details of prompt design for the Seeker micro-agent. In the system setup, {dataset_summary} is a placeholder for the current dataset summary user is working with. In the directive, the {petel}, and {context} are placeholders which will be replaced dynamically during the conversation

| Prompt Design |
|---|
| **System setup** |
| The AI assistant serves as a virtual data scientist, striving to complete a given JSON object by incorporating missing values through user input. Update the JSON object with the relevant unknown slot, considering that the user may also wish to modify existing filled slots. Ensure that the output is a strictly JSON format. If it is not possible to update the JSON object, return the original input JSON object. |
| **Demonstration** |
| Continued on next page |

**Table 8 – continued from previous page**

| | **Prompt Design** |
|---|---|
| **User** | Using the provided information (I'd like to forecast the total number of flight delays.), please update the appropriate slot in the given JSON object:
{
problem_type: time_series_forecasting,
target_variable: null,
forecast_horizon: null,
business_goals: null,
granularity: null,
features: null,
time_range: null,
model_preferences: null,
performance_metrics: null,
validation_method: null,
additional_requirements: null,
time_series_methods: null,
data_filters: [
column: null, condition: null, value: null,
column: null, condition: null, value: null
]
}
Ensure that you fill only the slot that is directly related to this information. Your response should consist solely of the updated JSON file. |
| **LLM** | {
problem_type: time_series_forecasting,
target_variables: [flight_delays],
forecast_horizon: null,
business_goals: null,
granularity: null,
features: null,
time_range: null,
model_preferences: null,
performance_metrics: null,
validation_method: null,
additional_requirements: null,
time_series_methods: null,
data_filters: [
column: null, condition: null, value: null,
column: null, condition: null, value: null
]
} |

**Table 8 – continued from previous page**

| | **Prompt Design** |
|---|---|
| **User** | Using the provided information (for time horizon I would like to use one month I guess.), please update the appropriate slot in the given JSON object:
{
problem_type: time_series_forecasting,
target_variables: [flight_delays],
forecast_horizon: null,
business_goals: null,
granularity: null,
features: null,
time_range: null,
model_preferences: null,
performance_metrics: null,
validation_method: null,
additional_requirements: null,
time_series_methods: null,
data_filters: [
column: null, condition: null, value: null,
column: null, condition: null, value: null
]
}
Ensure that you fill only the slot that is directly related to this information. Your response should consist solely of the updated JSON file. |
| **LLM** | {
problem_type: time_series_forecasting,
target_variables: [flight_delays],
forecast_horizon: 1 month,
business_goals: null,
granularity: null,
features: null,
time_range: null,
model_preferences: null,
performance_metrics: null,
validation_method: null,
additional_requirements: null,
time_series_methods: null,
data_filters: [
column: null, condition: null, value: null,
column: null, condition: null, value: null
]
} |

**Directive**

Using the provided information ({utterance}, {context}), please update the appropriate slot in the given JSON object: {petel}. Ensure that you fill only the slot that is directly related to this information. Your response should consist solely of the updated JSON file.

Table 8: The details of prompt design for the Feeder micro-agent. In the directive, the {utterance}, {context}, and {petel} are placeholders which will be replaced dynamically during the conversation

**Prompt Design**

**System setup**

**Table 9 – continued from previous page**

| Prompt Design |
|---|
| As the AI agent, your primary objective is to carefully examine the problem illustrated within the json object and then develop a comprehensive, yet clear, natural language description. This explanation should effectively communicate the underlying issue to the reader, ensuring that they can grasp the key aspects and implications of the problem at hand. By providing an elaborative and accessible interpretation, the AI agent will enable a better understanding of the situation for individuals who may not be familiar with the json format or the specific context of the issue. |

| Demonstration | |
|---|---|
| **User** | Please provide a clear and concise natural language description of the problem formulation presented here:
{
problem_type: classification,
target_variable: delay_severity,
features: [departure_airport, arrival_airport, airline, scheduled_departure_time, scheduled_arrival_time, weather_conditions],
dataset_size: 10000/Default,
performance_metrics: [accuracy, precision, recall, f1_score, confusion_matrix],
validation_method: cross_validation,
classification_methods: [logistic_regression, decision_tree_classifier, random_forest_classifier, svm_classifier, knn_classifier, xgboost_classifier, naive_bayes],
data_filters: [
column: delay_duration, condition: greater_than, value: 15,
column: departure_airport, condition: equals, value: JFK
],
business_goals: [reduce customer complaints, optimize scheduling, improve airport operations],
additional_requirements: [robust to outliers, handle class imbalance],
model_preferences: interpretable
} |
| **LLM** | This is a time series forecasting problem that aims to predict the total delay duration of flights for the next month. The prediction will be based on various features such as departure and arrival airports, airline, weather conditions, and delay duration. The model should be interpretable and capable of handling seasonality, trends, and nonlinear relationships. The problem also aims to understand the impact of weather on flight delays and to forecast delay duration per airline. The dataset is at hourly granularity and covers the past three years. The performance metrics used to evaluate the model include mean squared error, mean absolute error, and R2 score. The model will be validated using time series cross-validation. The data filters used in this problem include only flights departing from JFK airport and operated by Delta airlines. |

| Directive |
|---|
| Please provide a clear and concise natural language description of the problem formulation presented here: {petel}. |

Table 9: The details of prompt design for the PeTEL Descriptor micro-agent. In the directive, {petel} is a placeholders which will be replaced by a fully filled PeTEL expression.

## C Qualitative Examples

This section explores the qualitative performance of Large Language Models (LLMs) in complex tasks. We focus on ChatGPT as our chosen LLM and utilize the Student Performance dataset for our analysis. Our investigation revolves around three main areas. Firstly, we examine the chat cycle between users and the VIDS system, emphasizing the usability and efficacy of VIDS' LLMs in facilitating seamless conversations. Secondly, we explore the interaction among the micro-agents themselves, each equipped with its unique LLM instance. This analysis reveals the intricacies of micro-agent dynamics in orchestrating complex tasks. Lastly, we assess the versatility and resilience of the Prompt Taxonomy within these micro-agents, demonstrating the practicality and applicability of LLMs in demanding scenarios. Through this in-depth qualitative examination, we aim to provide a panoramic view of the essential role and effectiveness of LLMs in navigating complex tasks.

### C.1 Dataset

The Student Performance (SP) dataset, publicly accessible on Kaggle[3], includes 649 multilingual records with 33 diverse attributes. We use this dataset as a whole as if the user want to have some insight about this dataset (i.e. dataset not divided into training, testing, and validation splits also no data was excluded). Download the dataset directly from Kaggle, adhering to their usage terms and conditions. Since this dataset was pre-existing, there are no specific details pertaining to our data collection process or annotator instructions.

### C.2 Overall Chat Cycle

In this subsection, we focus on the essential dynamics of the user-system chat cycle, pivotal for effective communication within VIDS. Facilitated by a sophisticated ensemble of LLM agents, this cycle forms the backbone of user-system interactions. A deep dive into this cyclical process will elucidate how LLMs orchestrate fluid, efficient communication. This scrutiny will shine a light on the user experience, underscoring the prowess and utility of the LLMs in complex tasks. Table 10 presents a tangible snapshot of user-system interactions, exemplifying the successful execution of the user's goal task. The dialogue begins with the AI presenting a dataset summary and proposing potential ML tasks. The user selects clas-

sification as the most suitable task and identifies the 'final grade' as the target variable. The AI then proceeds to query about the size of the dataset, to which the user responds with '10,000 samples.' Following several exchanges (elided for brevity), the AI finalizes the task as predicting 'final grade' based on various features, considering three classification methods: random forest, SVM, and logistic regression. The data is filtered to include only records with attendance over 75% and study hours over 1. The primary goal is to predict student performance for early interventions, with model interpretability and accuracy considered vital requirements. The user then approves this task setup, closing this dialogue cycle.

### C.3 Interaction Between Micro-Agents

In this subsection, we delve into the interactions within the micro-agents themselves, each equipped with its own instance of a Large Language Model (LLM). These interactions occur in four distinct states within the system: data visualization, task formulation, prediction engineering, and result generation and summarization. By examining the interplay among these micro-agents in each state, we aim to enhance our understanding of the internal mechanics of the LLMs and their role in executing complex tasks.

### C.3.1 Data Visualization State:

The first state, data visualization, involves the presentation of data in a visual format that aids in understanding the complexities and patterns within the data. The interaction between micro-agent agents in this state is crucial as it determines the effectiveness of the data representation. In Table 11, we show the specific roles and interactions of the micro-agents in this state. The table presents a step-by-step interaction of different micro-agents involved in the Data Visualization state, utilizing a dataset related to student performance. The Dataset Summarizer initially provides a comprehensive summary of the dataset, including detailed descriptions of dataset content, column names, a sample row, and observed trends. Key insights, such as the correlation between a student's academic performance and factors like gender, study time, and alcohol consumption, are highlighted. The Suggest ML task micro-agent then proposes two potential machine learning tasks—Classification and Regression—that can be derived from the dataset. Classification could predict a student's success or failure, whereas Regression could estimate a student's final grade based on numerous factors, offering critical

---

[3]https://www.kaggle.com/datasets/larsen0966/student-performance-data-set

insights for intervention and performance enhancement strategies.

### C.3.2  Task Formulation State:

The second state, task formulation, is the stage where the task to be performed is defined and structured. The interaction between micro-agent agents in this state is critical as it shapes the formulation of the task and ultimately influences the success of the task's execution. In Table 12, we delve into the specifics of these interactions. The table illustrates a complex interaction of micro-agents within the Task Selection state, focusing on an example with a student performance dataset. The dialogue begins with a summary of the dataset and an initial classification task proposition by a user. As the conversation progresses, different agents come into play, summarizing the dialogue context, selecting states, feeding problem specifications, and asking questions to fill in missing data. The process is iterative, updating a JSON object specifying the problem and model preferences. This iterative dialogue concludes once all the necessary slots in the JSON object are filled, resulting in a comprehensive task description. This task involves predicting student final grades with a dataset of 10,000 samples and specific classification methods, filtering data based on attendance and study hours, and emphasizing the importance of model interpretability.

### C.3.3  Prediction Engineering State:

The third state, prediction engineering, is an integral part of the AutoML pipeline. As shown in Table 13, this state takes the formulated task and prepares the dataset accordingly. The interaction between micro-agent agents in this state is essential as it directly influences the preparation of the dataset, which in turn impacts the accuracy and reliability of the predictions made. In this state, the micro-agent agents work collaboratively to interpret the task requirements, adjust the dataset to align with these requirements, and set the stage for accurate prediction generation.

### C.3.4  Model Training, Result Summary, and Recommendation State:

After the task is formulated, VIDS interfaces with AutoML tools (e.g. AutoSKLearn) and trains downstream model(s) based on the task formulation determined beforehand. As shown in Table 14, from training performance of different models, VIDS generates summaries, including the results and recommendations based on user preferences defined in the task formulation. Our future work will be to interact with the user in this stage and evaluate different models based on the user's business goals.

| Turn | Conversation |
|---|---|
| $system_1$ | Welcome, I am your personal data scientist. I have processed the provided dataset for your convenience I present the summary here: {Summary}. I propose the following ML tasks for this dataset: {$ML\_task$} |
| $utterance_1$ | Ok, from the description it seems like classification is a good choice. |
| $response_1$ | Great! Let's move forward with the classification task. Now that we have decided on the classification task, let's move on to formulating the problem. Can you provide me with more details on what you would like to achieve with this task? For example, what is the target variable you would like to predict? |
| $utterance_2$ | I think I will use final grade |
| $response_2$ | Based on the provided context, the next unidentified slot in the JSON object is 'dataset_size'. To obtain this information, the agent could ask the user: Can you please provide the size of your dataset for the Classification model you are building? |
| $utterance_3$ | Lets use 10000 samples for this task |
| | ... ... ... |
| $response_n$ | The task is to predict the final_grade of a student based on various features like study_hours, attendance, participation, homework_scores, and test_scores. The dataset size is hypothetical. Performance is measured using accuracy, F1 score, and the confusion matrix. K-fold cross-validation is used for validation. Three classification methods are considered: random forest, SVM, and logistic regression. The data is filtered to only include records where attendance is greater than 75 (assuming attendance is measured as a percentage) and study_hours is more than 1. The business goal is to predict student performance to implement early interventions for those at risk of failing. Model interpretability is highlighted as an important additional requirement, and there's a preference for models with higher accuracy, but interpretability is also considered important. |
| $utterance_{n+1}$ | That seems all right to me. go ahead with this task. |

Table 10: Interaction of user and system where $utterance_*$ are the user utterances and $response_*$ are the overall system response to the user. In $system_1$, {summary} and {$ML\_task$} are from Table 11.

| # | Input | Micro-Agent | Output |
|---|---|---|---|
| 1 | Dataset | — | $Min\_ds$ : Miniature dataset |

Continued on next page

**Table 11 – continued from previous page**

| # | Input | Micro-Agent | Output |
|---|-------|-------------|--------|
| 2 | *Min_ds* | Dataset Summarizer | *summary* : { |
| | | | dataset_summary : This dataset contains information about students in a school. It includes their demographic information such as sex, age, address, family size, and parents' education level. It also includes information about their academic performance such as grades, study time, and number of failures. Additionally, it includes information about their social life such as alcohol consumption, going out with friends, and having a romantic relationship. |
| | | | columns : list of column names and description |
| | | | sample row : This row represents a female student who is 16 years old, lives in an urban area with a family size greater than 3, and her parents are living together. Her mother has a higher education level than her father and works in the health sector, while her father works in the services sector. She chose this school because of its reputation and her father is her guardian. It takes her less than 15 minutes to travel from home to school. She studies for 2 to 5 hours per week and has not failed any classes in the past. She receives no extra educational support from school or family members, and does not pay for extra classes. She participates in extracurricular activities and attended nursery school. She wants to take higher education and has internet access at home. She is not in a romantic relationship and has a good quality of family relationships. She has a moderate amount of free time and goes out with friends occasionally. She has a low workday alcohol consumption and a very low weekend alcohol consumption. Her current health status is good and she has had 6 school absences. Her grades are 12 in the first period, 12 in the second period, and 13 in the final period. |
| | | | trend : From the dataset, it can be observed that female students tend to have higher grades than male students. Additionally, students who receive extra educational support from school or family members tend to have higher grades than those who do not. Students who have a higher amount of study time per week also tend to have higher grades. On the other hand, students who have a higher amount of alcohol consumption tend to have lower grades. Finally, students who have a higher number of school absences tend to have lower grades. } |
| 3 | *summary* | Task Suggestor | **ML_task :** Based on the provided dataset summary, the following are the suitable ML tasks: |
| | | | 1. Classification: Classification can be used to predict whether a student will pass or fail based on their demographic information, academic performance, and social life. This task can be formulated as a binary classification problem where the target variable is pass/fail. The features can include sex, age, address, family size, parents' education level, study time, number of failures, alcohol consumption, and school absences. |
| | | | 2. Regression: Regression can be used to predict the final grade of a student based on their demographic information, academic performance, and social life. This task can be formulated as a regression problem where the target variable is the final grade (G3). The features can include sex, age, address, family size, parents' education level, study time, number of failures, alcohol consumption, and school absences. |
| | | | The rationale for choosing classification is that it can help identify students who are at risk of failing and provide early intervention to improve their academic performance. The rationale for choosing regression is that it can help identify factors that contribute to a student's academic success and provide insights into how to improve their performance. |

Table 11: Interaction of different micro-agent in the Data Visualization state. In this qualitative study we work with Flight Delay dataset.

| # | Input | Micro-Agent | Output |
|---|-------|-------------|--------|
| 1 | $utterance_1$, $summary$, $ML\_task$ | Dialogue summarizer | $context_1$ : The dialogue provides a summary of a dataset containing information about students in a school, including their demographic information, academic performance, and social life. The dataset summary highlights trends such as female students having higher grades and students who receive extra educational support having higher grades. Based on this information, the user suggests that classification would be a suitable machine learning task to predict whether a student will pass or fail based on their demographic information, academic performance, and social life. The user's rationale for choosing classification is to identify students who are at risk of failing and provide early intervention to improve their academic performance. |
| 2 | $context_1$, $utterance_1$ | State Detector | $ds_1$ : Task Formulation |
| 3 | $utterance_1$, $context_1$, $ds_1$ | Task Selector | $ML\_task$ : { "model": "classification", "reason": "Based on the dataset summary, the AI agent suggested two suitable machine learning tasks: classification to predict whether a student will pass or fail, and regression to predict a student's final grade. The user agreed that classification would be a good choice based on the dataset summary. Therefore, a classification model is a good choice for this task."} |
| 4 | $utterance_2$, $context_1$, $ds_1$ | Dialogue Summarizer | $context_2$ : In this dialogue, the AI assistant provides a summary of a dataset containing information about students in a school, including their demographic information, academic performance, and social life. The assistant notes that female students tend to have higher grades than male students, and students who receive extra educational support from school or family members tend to have higher grades. The assistant suggests two suitable ML tasks for the dataset: classification and regression. The user decides to use classification to predict whether a student will pass or fail based on their demographic information, academic performance, and social life. The assistant asks for more details on the target variable, and the user decides to use the final grade as the target variable. |
| 5 | $context_2$, $utterance_2$ | State Detector | $ds_2$ : Task Formulation |
| 6 | $utterance_2$, $context_2$, $PeTEL_1$ | Feeder | $PeTEL_2$ :

```
{
    problem_type: classification,
    target_variable: final grade,
    features: None,
    dataset_size: None,
    performance_metrics: None,
    validation_method: None,
    classification_methods: None,
    data_filters: [
        {column: final grade, condition: less than
            , value: passing grade},
        {column: None, condition: None, value:
            None}
        ],
    business_goals: None,
    additional_requirements: None,
    model_preferences: None

}
``` |

**Table 12 – continued from previous page**

| # | Input | Micro-Agent | Output |
|---|-------|-------------|--------|
| 7 | $utterance_2$, $context_2$, $PeTEL_2$ | Seeker | $Q_1$ : Based on the provided context, the next unidentified slot in the JSON object is 'dataset_size'. To obtain this information, the agent could ask the user: Can you please provide the size of your dataset for the Classification model you are building? |
| 8 | $utterance_3$, $context_2$, $Q_1$ | Dialogue Summarizer | $context_3$ : The AI assistant provided a summary of the dataset, including information about the columns and a sample row. The assistant also mentioned trends observed in the dataset, such as the correlation between study time and grades, and the negative correlation between alcohol consumption and grades. The user asked for suggestions on machine learning tasks, and the assistant suggested classification and regression. The user chose classification and specified that the target attribute would be the final grade. The assistant then asked for more information about the problem to help define it more precisely. The user decided to use 10,000 samples for the task |
| 9 | $utterance_3$, $context_3$, $PeTEL_3$ | Feeder | $PeTEL_3$ :

```
{
    problem_type: classification,
    target_variable: final grade,
    features: None,
    dataset_size: 10000,
    performance_metrics: None,
    validation_method: None,
    classification_methods: None,
    data_filters: [
        {column: final grade, condition: less than
            , value: passing grade},
        {column: None, condition: None, value:
            None}
        ],
    business_goals: None,
    additional_requirements: None,
    model_preferences: None

}
``` |
| 10 | $PeTEL_3$ | PeTEL Checker | $PeTEL\_is\_complete$ |
| | Repeat step 6 - 10 until PeTEL is complete. Completed PeTEL is denoted by: $PeTEL^*$ | | |
| n | $PeTEL^*$, $context_n$ | PeTEL Descriptor | $Description$ :
The task is to predict the final_grade of a student based on various features like study_hours, attendance, participation, homework_scores, and test_scores. The dataset size is hypothetical. Performance is measured using accuracy, F1 score, and the confusion matrix. K-fold cross-validation is used for validation. Three classification methods are considered: random forest, SVM, and logistic regression.
The data is filtered to only include records where attendance is greater than 75 (assuming attendance is measured as a percentage) and study_hours is more than 1. The business goal is to predict student performance to implement early interventions for those at risk of failing. Model interpretability is highlighted as an important additional requirement, and there's a preference for models with higher accuracy, but interpretability is also considered important. |

**Table 12 – continued from previous page**

| # | Input | Micro-Agent | Output |
|---|-------|-------------|--------|
| n+1 | $context_{n+1}$, $utterance_{n+1}$ | State Detector | $ds_{n+1}$ : Prediction Engineering |

Table 12: Interaction of different micro-agent in the Task Selection state. In the table, $utterance_1, utterance_2, utterance_3$ are from Table 10, and $summary, ML\_task$ are from Table 11. At 10th row, Micro-Agent PeTEL Checker is a python function which check existance of null value in PeTEL.

| # | Input | Micro-Agent | Output |
|---|-------|-------------|--------|
| 1 | $PeTEL^*$ | PeTEL to Feature Converter | List of attributes |
| 2 | $PeTEL^*$ | Data Preparer | Prepares data with the conditions in PeTEL |
| 3 | $PeTEL^*$ | AutoML Interfacer | Calls the AutoML interface |

Table 13: Interaction of different micro-agent in the Prediction Engineering state.

| Step | Input | Micro-Agent | Output |
|------|-------|-------------|--------|
| 1 | $context_n$, $PeTEL^*$, AutoML response | Result Summarizer | $Result$ : performance of each model based on evaluation criteria set in problem formulation. |
| 2 | $context_n$, $Result$ | Result Visualizer | $Output$ : Description of results in natural language. |

Table 14: Interaction of different micro-agent in the Task Formulation state. In the table, $utterance_1, utterance_2, utterance_3$ are from Table 10

## C.4 Prompt Engineering Taxonomy

The successful collaboration between humans and artificial intelligence in complex tasks necessitates a comprehensive understanding of the various levels of interaction that occur between them. These levels span from Level 0, where AI is solely responsible for data processing, to Level 5, which involves the integration of evaluation criteria. Building upon the foundational work on taxonomy of prompt engineering (TELeR) by Santu & Feng (2023), we put forward the notion of considering the depth of information that the System Role discloses to the Large Language Model (LLM). To illustrate, if a system role is well-delineated, it precludes its prompt from being classified as Level 0. This study will specifically focus on three micro-agents: the Intent and State Detector, the Dialogue Summarizer, and the Conversation Manager. Each of these micro-agents plays a unique and integral role in fostering a dynamic and functional dialogue between the user and the AI, leading to a more streamlined and efficient system overall. The revised taxonomy for these interaction levels is as follows:

**Level 0**: No directive is given. The focus is solely on the exchange of data.

**Level 1**: A simple one-sentence directive is provided, expressing the high-level goal of the task.

**Level 2**: A multi-sentence (paragraph-style) directive is given, expressing the high-level goal and the sub-tasks needed to achieve this goal.

**Level 3**: A complex directive is provided, expressing the high-level goal along with a bulleted list of subtasks that need to be performed.

**Level 4**: This level includes a complex directive that encompasses the following: 1) A description of the high-level goal, 2) A detailed bulleted list of subtasks, and 3) An explicit statement asking the LLM to explain its response.

**Level 5**: This level includes a complex directive that encompasses the following: 1) A description of the high-level goal, 2) A detailed bulleted list of subtasks, 3) An explicit statement asking the LLM to explain its response, and 4) A guideline about how the response should be evaluated.

By understanding these levels of interaction, we can maximize the potential benefits of AI and guide future research into user experience, system performance, and ethical considerations in AI applications.

### C.4.1 Intent and State Detector micro-agent:

In this case study, each level of the prompt will be constructed based on the following input:

1. **context:** The user and the AI assistant discussed the summary of a dataset containing information about students in a school, including their demographic information, academic performance, and social life. The AI assistant suggested two suitable machine learning tasks based on the dataset: classification and regression. The user agreed that classification is a good choice to identify students who are at risk of failing and provide early intervention to improve their academic performance.

2. **state:** dataset_understanding

3. **utterance:** Ok, from the description it seems like classification is a good choice.

By following the prompt taxonomy we discussed earlier, Table 15 represents the response from the LLM agent for each level of prompt.

### C.4.2 Dialogue Summarizer micro-agent:

In this case study, each level of the prompt will be constructed based on the following history as its input:

- **history:**
  1. assistant: summary of Student Performance dataset
  2. user: From the summary you provided can you suggest me any machine learning task?
  3. assistant: $context_1$ from Table 12
  4. user: Ok, from the description it seems like classification is a good choice.

By following the prompt taxonomy we discussed earlier, Table 16 represents the response from the LLM agent for each level of prompt.

### C.4.3 Conversation Manager micro-agent

In this case study, each level of the prompt will be constructed based on the following input:

1. **context:** The user and assistant discussed the dataset containing information about

students in a school, including their demographic information, academic performance, and social life. The assistant suggested that the suitable machine learning tasks for this dataset are classification and regression, with classification being a good choice to identify students at risk of failing and provide early intervention to improve their academic performance. The user agreed that classification is a good choice.

2. **state:** Dataset Visualization

3. **utterance:** Ok, from the description it seems like classification is a good choice.

4. **intent:** Select problem

5. **mp_resp:** {"model": "classification", "reason": "Classification is a good choice for identifying students at risk of failing and providing early intervention to improve their academic performance. The dataset contains demographic information, academic performance, and social life, which can be used as input features to predict the likelihood of a student failing or passing."}

By following the prompt taxonomy we discussed earlier, Table 17 represents the response from the LLM agent for each level of prompt.

| Prompt Level | Response From LLM |
|---|---|
| 0 | Can you tell me more about the dataset? What are the features and labels? |
| 1 | Intent: dataset_understanding State: The user and the AI assistant have discussed a dataset containing information about students in a school, and the AI assistant has suggested two suitable machine learning tasks: classification and regression. The user has agreed that classification is a good choice to identify students who are at risk of failing and provide early intervention to improve their academic performance. |
| 2 | { intent: Select problem, current_state: Dataset_Understanding, next_state: Task_Formulation } |
| 3 | { intent: Problem Execution, current_state: Dataset_Understanding, next_state: Task_Formulation } |
| 4 | {intent: Select problem, current_state: dataset_understanding, next_state: Task_Formulation, explanation: The user has indicated that they understand the dataset and have chosen classification as a suitable problem. Therefore, the next state should be problem_selection, where the user can select a problem related to classification.} |
| 5 | {intent: Select problem, current_state: dataset_understanding, next_state: problem_selection, explanation: The user has acknowledged the suggestion of classification as a suitable problem and is now ready to move to problem selection state. The dataset understanding state has been completed, and problem selection is the next logical step in the conversation flow., evaluation: The agent has correctly identified the user's intent and provided a clear explanation for the state transition. The evaluation criteria for the agent's response are: accuracy, clarity, and relevance to the conversation context.} |

Table 15: State Detector micro-agent response for different level of prompt engineering input.

| Prompt Level | Response From LLM |
|---|---|
| 0 | Great! Classification is a suitable choice for predicting whether a student will pass or fail based on their demographic information, academic performance, and social life. You can use various classification algorithms such as logistic regression, decision trees, random forests, and support vector machines to build a model that can accurately predict the target variable. Additionally, you can use techniques such as feature selection and hyperparameter tuning to improve the performance of the model. |
| 1 | Great! Classification is a suitable choice for predicting whether a student will pass or fail based on their demographic information, academic performance, and social life. This can help identify students who are at risk of failing and provide early intervention to improve their academic performance. |
| 2 | The user and assistant discussed the summary of a dataset containing information about students in a school, including their demographic information, academic performance, and social life. The assistant suggested two suitable machine learning tasks based on the dataset: classification and regression. The user decided that classification would be a good choice to identify students at risk of failing and provide early intervention. |
| 3 | Based on the dataset summary, classification is indeed a suitable machine learning task to predict whether a student will pass or fail based on their demographic information, academic performance, and social life. This can be formulated as a binary classification problem where the target variable is pass/fail and the features can include sex, age, address, family size, parents' education level, study time, number of failures, alcohol consumption, and school absences. |
| 4 | Based on the dataset summary, classification is indeed a suitable choice for predicting whether a student will pass or fail based on their demographic information, academic performance, and social life. This task can help identify students who are at risk of failing and provide early intervention to improve their academic performance. |

**Table 16 – continued from previous page**

| Prompt Level | Response From LLM |
|---|---|
| 5 | Based on the dataset summary, it does seem like classification is a suitable machine learning task. It can help identify students who are at risk of failing and provide early intervention to improve their academic performance. The features that can be used for classification include sex, age, address, family size, parents' education level, study time, number of failures, alcohol consumption, and school absences. |

Table 16: Dialogue Summarizer micro-agent response for different level of prompt engineering input.

| Prompt Level | Response From LLM |
|---|---|
| 0 | The assistant then asked the user about the specific features in the dataset that could be used for classification. The user mentioned that features such as gender, age, ethnicity, parental education level, and previous academic performance could be used to predict whether a student is at risk of failing. The assistant agreed and suggested that the dataset could be split into training and testing sets to build and evaluate a classification model. The user agreed and asked the assistant for recommendations on which classification algorithms to use. The assistant suggested using logistic regression, decision trees, and random forests, as they are commonly used for classification tasks and can handle both categorical and numerical data. The user thanked the assistant for the suggestions and said they would explore these algorithms further. |
| 1 | Yes, based on the information you provided, classification seems like a suitable machine learning task for the dataset containing information about students in a school. It can help identify students who are at risk of failing and provide early intervention to improve their academic performance. Regression can also be a good choice to predict the academic performance of students based on their demographic and social information. |
| 2 | Great! Now that we have agreed that classification is a suitable task for your dataset, let's move on to formulating the problem. Can you please provide me with more details about the specific problem you want to solve using this dataset? For example, do you want to predict which students are at risk of failing a particular subject or identify factors that contribute to academic success? This information will help us formulate a more precise ML task that meets your needs. |
| 3 | Based on our conversation so far, it seems like classification is a suitable model for identifying students who are at risk of failing and providing early intervention to improve their academic performance. By analyzing various factors such as attendance, grades, and behavior, a classification model can predict which students are likely to fail and provide targeted support to improve their academic outcomes. Now that we have selected the ML task, let's move on to formulating the problem. Can you provide more information about the dataset you have and the specific problem you want to solve with this classification model? This will help us to formulate the problem and select appropriate parameters for the model. |
| 4 | Based on the given context, it seems like we have reached a consensus that classification is a suitable machine learning task for the dataset containing information about students in a school. Now, we need to formulate the problem to proceed further. To do that, I suggest we move to the next conversation state, which is "problem_formulation". In this state, we can discuss the specific details of the classification task, such as the target variable, the features to be used, and the algorithm to be employed. So, to move forward, can you please provide me with more information about the problem you want to solve using classification? This will help me formulate the problem and suggest relevant solutions to you. |

Continued on next page

**Table 17 – continued from previous page**

| Prompt Level | Response From LLM |
| --- | --- |
| 5 | Based on the current conversation state, it seems like we are in the Task_Formulation state, and the user has agreed that classification is a good choice for the given dataset. Therefore, we can proceed with formulating the classification problem. |
| | To do so, we need to determine the target variable that we want to predict. In this case, we want to identify students who are at risk of failing and provide early intervention to improve their academic performance. Therefore, we can use the "final grade" variable as our target variable, and classify students into two categories: those who are at risk of failing and those who are not. |
| | To achieve this, we can use a binary classification algorithm, such as logistic regression or decision tree. We can also use feature selection techniques to identify the most important features that contribute to the prediction of the target variable. |
| | Based on this, we can formulate the ML task as follows: Develop a binary classification model to predict whether a student is at risk of failing based on their demographic information, academic performance, and social life. The target variable is the final grade, and the model should be trained on a labeled dataset. Feature selection techniques can be used to identify the most important features. |
| | This task can be further refined based on the users̀ specific requirements and preferences. For example, the user may want to use a different classification algorithm or include additional features in the model. |
| | Evaluation criteria for the generated response: |
| | - The response provides a clear and concise formulation of the ML task based on the users̀ requirements and preferences. |
| | - The response takes into account the dataset description and the previous discussion between the user and the assistant. |
| | - The response suggests an appropriate classification algorithm and feature selection techniques based on the problem requirements. |
| | - The response is open-ended and allows for further refinement and customization based on the users̀ specific needs |

Table 17: Conversation Manager micro-agent response for different level
of prompt engineering input.

## D   Reproducibility Criteria

For all reported experimental results:

- A clear description of the mathematical setting, algorithm, and/or model.

  – Model description is provided in Section 3.

- Submission of a zip file containing source code, with specification of all dependencies, including external libraries, or a link to such resources (while still anonymized) Description of computing infrastructure used.

  – Zip file containing source code will be submitted in submission as supplementary material.

- The average runtime for each model or algorithm (e.g., training, inference, etc.), or estimated energy cost

  – We used ChatGPT which provides API for getting response hence we are unable to estimate such performance.

- Number of parameters in each model

  – We use ChatGPT as the LLM in our case study. As we do not get any such information from any document of ChatGPT, we can not report such number.

- Corresponding validation performance for each reported test result

  – We utilize the LLM (ChatGPT) through API, we do not train or finetune the model hence we can not report such performance metric.

- Explanation of evaluation metrics used, with links to code

  – We provide qualitative study in this paper i.e. we depend on human judgment for the evaluation.

For all experiments with hyperparameter search:

- The exact number of training and evaluation runs

  – We utilize the LLM (ChatGPT) through API, we do not train or finetune the model hence we can not report such number.

- Bounds for each hyperparameter

  – We utilize the LLM (ChatGPT) through API, we do not train or finetune the model hence we can not report such number.

- Hyperparameter configurations for best-performing models

  – We utilize the LLM (ChatGPT) through API, we do not train or finetune the model hence we can not report such number.

- Number of hyperparameter search trials

  – We utilize the LLM (ChatGPT) through API, we do not train or finetune the model hence we can not report such number.

- The method of choosing hyperparameter values (e.g., uniform sampling, manual tuning, etc.) and the criterion used to select among them (e.g., accuracy)

  – We utilize the LLM (ChatGPT) through API, we do not train or finetune the model hence we can not report such number.

- Summary statistics of the results (e.g., mean, variance, error bars, etc.)

  – We utilize the LLM (ChatGPT) through API, we do not train or finetune the model hence we can not report such number.

For all datasets used:

- Relevant details such as languages, and number of examples and label distributions

  – Refer to Section C.1.

- Details of train/validation/test splits

  – Refer to Section C.1.

- Explanation of any data that were excluded, and all pre-processing steps

  – Refer to Section C.1.

- A zip file containing data or link to a downloadable version of the data

  – Refer to Section C.1.

- For new data collected, a complete description of the data collection process, such as instructions to annotators and methods for quality control.

  – Refer to Section C.1.

