# OpenReview forum: "Large Language Models as your Personal Data Scientist"
_TMLR — Rejected by TMLR_

### Review · Reviewer_tDKy · 2023-10-05

**Summary Of Contributions:**

The paper deals with applying LLMs to an interesting area: providing a natural language conversational interface to systems like AutoML used for common data science tasks. The paper use a LangChain-like approach to:
1) define the various components needed to accomplish the data science task: Data visualization, Task Formulation, Prediction engineering, Results summarization and recommendation.
2) connect various components required to accomplish a data science task by
 * iteratively invoking the LLM
 * refining the prompt to account for the state, task objective
 * performing slot-filling where necessary to augment the prompt with needed info.
3) create separate global components for 2) above so that the whole framework can operate in the context of a conversational assistant.
4) ultimately generate an execution plan that can be run on the user provided dataset in AutoML.

These components and their subcomponents are termed as micro-agents in the paper.

The paper details one possible design of such a system in it's entirety, named Virtual Interactive Data Scientist (VIDS). It provides the contents of the prompts used in each of the components. It also discusses the qualitative effect of adding more information in the prompt

**Audience:**

Yes

**Claims And Evidence:**

No

**Requested Changes:**

* The paper mentions the notion of ill-defined tasks, but presents examples where the tasks are from a relatively well-defined set. What is an example of this ill-defined task (since the example from Table 10 seems well-defined)? Is there a prior step missing where the user initiates the conversation with an overall goal (e.g. "I want to find out what I can and cannot predict reliably from my dataset")?
* Performing Data Science on a given dataset is complex for practitioners not well-versed with the field. The paper postulates that a conversational AI or a dialog based interface is a feasible and preferred way to facilitate this. Can you please point to any prior evaluation-based study to support this? (i.e. If it can be shown that an end-to-end opinionated system - which only provides the users fixed levers to update the model - is inferior to a more free-form dialog based system (not necessarily LLM based))
* In Section 2: "...the true upper limit of LLMs remains a tantalizing unknown..." which domains are the limits being mentioned here about?
* In Section 3: "...This stateless design ensures a smooth narrative flow and avoids complications of state-dependent biases or entanglements, thus bolstering the versatility and adaptability of our dialogue system..." Can you please elaborate/provide example or reference on what would be a smooth vs non-smooth narrative flow? Also, can you please add some examples of entanglement-related complications? Additionally, can you please state what exactly is meant by stateless here (since based on the examples in Tables 1, 2, 3 the state and context of the conversation so far is passed into the prompt in every invocation).
* In Section 3.2.1: "...These cooperative micro-agents ensure efficient dataset exploration and readiness for the Task Formulation phase..." Can you please explain what is meant by cooperative here? Based on Figure 2 and description in Section 3.2, the outputs from Dataset Summarizer and Task Suggestor micro-agents are then fed into Task formulation, but they don't affect each other.
* Significant portions of Section 3.5 mentions about proposed future work for this component. It would probably be better to move this into the Future work or Discussions section.
* Section 4: "...demonstrating the practicality and applicability of LLMs in demanding scenarios..." Can you please add some specificity to the scenarios? Are they demanding because of need to do CoT, or the length of the prompt is too long (hence complex)
* Most of the information in Section 4.2 has previously been mentioned in the description of the micro-agents. It does not provide any information about the interaction, hence feels redundant. Consider updating?
* In Section 5.1 Strengths of Dialogue Flow Control: "LLM (ChatGPT) performed commendably in terms of overall dialogue flow control by carefully balancing multiple factors, such as the utterance, context, responses from various microagents, user intent, and the current state of the conversation." Based on the prior sections, the LLMs were individual global and local micro-agents within a (presumably) deterministic overall dialog workflow. Is there another root LLM agent that decides which of the global or local micro-agent to run? If yes, could you please add details about that? If not, could you please explain how it is the LLM that does the overall dialogue flow control?
* The Appendix sections listing the micro-agents do not add any new information from what is presented in the main section of the paper. For instance, the Dataset Summarizer has no details on how the subset of data is selected by the user to embed in the prompt for that micro-agent to use for summarization. Please consider adding in more/new detail in these sections or merge the existing appendix sections into the main paper.
* Typo on pg 20: "ask selection" -> "task selection"
* Section C.4 mentions the paper used a revised version of the TELeR taxonomy. But the version in that section is the same as the version in the reference. Can you please update? Also, can you please provide a clear example of how those levels translates to different *input* prompts for the various micro-agents?
* The temperature setting for the OpenAI calls have not been specified, nor the randomizer's seed. Can you please mention either of those for reproducibility?

**Strengths And Weaknesses:**

Strengths
========
1) Explores a very interesting, nascent and impactful area of research - using LLM advances for democratization of data science.
2) References quite relevant literature for a detailed systems-level approach to solving a data science problem.
3) The paper's main strength is clearly designing the VIDS sytem with various components - each with a capability granular enough that it can be accomplished by straightforward prompting techniques in LLMs (like LangChain)
4) Able to demonstrate successfully completing an AutoML task via dialog (that might otherwise require complex UI navigation).

Weaknesses
==========
1) I'm not sure if the paper clearly showed that conversational AI is feasible or preferable way to perform data science tasks in general. The paper would have to include examples of more query or analysis based tasks and leverage the reasoning capabilities of the LLM (vs leveraging it's tool-use capabilities as in the current paper).
2) The paper doesn't answer the question about the accuracy of LLMs in framing and solving ill-defined
complex data science tasks. The paper has qualitative analysis on a single dataset and a single task type. To ascertain accuracy, precision, or efficacy, it would need at least more than one example. It needs a benchmark to compare against, analysis of whether the output of the micro-agents themselves are accurate or not (and preferable an ablation analysis). Even if doing a qualitative analysis, it needs review by human raters to see if the outputs, summaries, solutions were usable and accurate, even if successfully completing the entire dialog workflow.
3) The paper does not list details of why the specific dataset was chosen, how the mini version of that was selected so that the summarizer can extract trends.
4) When discussing the strengths of the approach in Section 5.1, the paper does not list any metrics or criteria for evaluating precision control or efficiency. So it is hard to gauge whether the performance or the strength of the approach.

---

> ### Author Response · Authors · 2023-10-12
> **Discussion and clarification of reviewer concerns**
>
> Thank you for your detailed feedback and insights on our manuscript. We value your comments and are committed to addressing each of your concerns.
>
> ## Inclusion of User Study and Quantitative Analysis
>
> Addressing the concern raised in the first, second, and fourth points of weakness:
>
> We acknowledge the concerns raised regarding the feasibility of conversational AI for data science tasks, the accuracy of LLMs in complex tasks, and the evaluation metrics in Section 5.1. To address these, we'd like to highlight that we have initiated a comprehensive user study, which is currently underway. This study involves a diverse group of participants, encompassing both data science experts and novices. We anticipate its completion within the next month.
>
> For accuracy, we have already performed an exhaustive quantitative analysis. Due to space constraints, we had to make the difficult decision of splitting this work into two papers, one paper focusing on the architecture and the other paper focusing on the quantitative analysis. The quantitative analysis paper is also under review at TMLR and can be found at this anonymized link: https://openreview.net/forum?id=ECVG0N8-Hs
>
> To further clarify, the primary objective of this paper is to delineate the architecture of an actual end-to-end conversational data science pipeline. In contrast, the paper we've separately submitted to TMLR centers on a quantitative study of the VIDS agent's prediction accuracies. Given the large amounts of results we wanted to share with the community, we decided to present these facets in two distinct submissions for clarity. However, if integrating the quantitative findings from the aforementioned paper would enrich the current manuscript, we are more than willing to do so. Please let us know, and we will merge the two submissions (though it will be a little challenging).
>
> The insights and findings from this user study and quantitative results will provide a clearer picture of the accuracy, precision, and efficacy of our approach. Moreover, it will offer a more objective assessment of the strengths and feasibility of our conversational AI system for data science tasks.
>
>   We are committed to integrating these findings into our revised manuscript, ensuring a more comprehensive and evidence-backed presentation of our work. Please advise us in terms of space management with all the results.
>
> ## Dataset Selection and Versatility of VIDS Agent
>
> Addressing the concern raised in the third point in weakness regarding the specificity of the dataset selection:
> The quantitative analysis paper at this anonymized link: https://openreview.net/forum?id=ECVG0N8-Hs includes VIDS’s performance on three different data sets: Online Delivery, Student Performance, and Flight Delay datasets. So, we have already tested VIDS on multiple data sets to ensure generality. Having said that, the foundational idea of this paper is to demonstrate an end-to-end architecture that can adapt and converse with users regardless of the dataset they upload. The intention is to showcase the agent's capability to handle any dataset on the fly and collaboratively formulate an ML task with the user. The dataset presented in the experiment was chosen randomly to exemplify this adaptability. We understand that this was not clearly communicated in our initial manuscript, and we will make the necessary revisions to elucidate this design choice and rationale. Moreover, during our user study, we plan to use a pool of more than *ten* datasets to measure the versatility of VIDS at the end-user level as well.

---

> > ### Author Response · Authors · 2023-10-12
> > **Continuation of Discussion and clarification of reviewer concerns**
> >
> > ## Clarification on ill-defined task
> >
> > In response to your query about the notion of "ill-defined tasks":
> >
> > An ill-defined task is characterized by its lack of specific parameters, ambiguous goals, multiple potential solutions, and the absence of a clear evaluation criterion. The process of ML task formulation can be aptly categorized as an ill-defined task. This is because the formulation is contingent upon various factors such as user preferences, dataset suitability, and other nuanced criteria.
> >
> > Table 10 provides a bird's-eye view of the model's interaction. To understand the depth of the interaction, it's essential to consider Tables 10 to 14 in tandem. For instance, in System1 and utterance1, the VIDS agent first offers a dataset summary, followed by ML task recommendations based on that summary. The user's choice, in this case, was a classification task. However, this is just one of the myriad possibilities. Depending on user decisions, numerous variations of ML tasks may emerge.
> >
> > For a comprehensive understanding of the VIDS agent's inner workings, it's crucial to view Tables 10 through 14 collectively. Due to space constraints, we presented the information in this segmented manner, but viewing these tables together provides a holistic picture of the agent's capabilities and interactions.
> >
> > We will try clarifying this in our revised manuscript to ensure readers can seamlessly navigate and understand the presented content.
> >
> > In terms of numbers, the quantitative analysis paper at this anonymized link: https://openreview.net/forum?id=ECVG0N8-Hs includes VIDS’s performance on formalizing ill-defined tasks with provided with partial information.
> >
> > ## Include References
> >
> > Addressing the following change request of including prior study to support the claim - "Performing Data Science on a given dataset is complex for practitioners not well-versed with the field. A conversational AI or a dialog-based interface is a feasible and preferred way to facilitate this.":
> >
> > Below are articles that discuss how Conversational AI is helpful for complex tasks in general [1] and data science (more specifically) [2,3]:
> >
> > 1. Fast E, Chen B, Mendelsohn J, Bassen J, Bernstein MS. Iris: A conversational agent for complex tasks. InProceedings of the 2018 CHI conference on human factors in computing systems 2018 Apr 21 (pp. 1-12).
> > 2. Pinoli P, Crovari P, Ieva F, Garzotto F, Ceri S. Ask Your Data-Supporting Data Science Processes by Combining AutoML and Conversational Interfaces. IEEE Access. 2023 May 2.
> > 3. Karmaker SK, Hassan MM, Smith MJ, Xu L, Zhai C, Veeramachaneni K. Automl to date and beyond: Challenges and opportunities. ACM Computing Surveys (CSUR). 2021 Oct 4;54(8):1-36.
> > We will cite these papers in the updated manuscript.
> >
> >
> > ## Domains of LLM Limits in Section 2
> >
> > LLMs have been evaluated across various well-defined tasks, and their performance in these tasks is commendable. However, to truly gauge the capabilities of LLMs, it's essential to test them in more complex and ill-defined tasks. Such evaluations could unveil the upper limits of LLMs. Our manuscript emphasizes this perspective, and we will ensure that this point is articulated more clearly in the revised version.

---

> > > ### Author Response · Authors · 2023-10-12
> > > **Continuation (2) of Discussion and clarification of reviewer concerns**
> > >
> > > ## Stateless Design and Smooth Narrative Flow in Section 3
> > >
> > > A smooth narrative flow in this context means that the AI understands user inputs and provides relevant, logical responses, maintaining the context throughout the conversation. A non-smooth flow would be when the AI provides disjointed or irrelevant responses, loses track of the conversation's context, or fails to maintain a logical progression. Each of our global agents performs a specific task to control overall narrative flow. The Dialog Summarizer ensures that only the most relevant information is passed to local agents, reducing noise and ensuring clarity. With the State Detector identifying the conversation's current state, local agents can adapt their responses to fit the context, ensuring relevance. The Conversation Manager ensures that the conversation progresses logically and coherently, guiding local agents and ensuring smooth transitions. At every interaction, the “context” is the response from the “Dialog Summarizer” and “state” is the response from the “State Detector”. Given the user input and previous dialog summary, those two global agents determine the current “context” and “state” of the dialogue.
> > >
> > > What we mean by “stateless” here is that there are two types of micro-agents: global and local. Global micro-agents are independent of the current state, while local micro-agents are directly dependent on the same. We understand that the name “stateless” is misleading, and we will change this term from “stateless” to “state-independent”. The proposed VIDS architecture ensures a smooth dialog flow via a delicate balance and coordination among the state-independent global micro-agents and state-dependent local micro-agents. More specifically, the local micro-agents concentrate on specific states, which ensure minimal entanglement with previous states. In contrast, global agents prioritize overseeing the overall dialogue flow. This dual focus ensures a streamlined conversation, with each agent playing its part to maintain clarity and coherence.
> > >
> > >
> > > ## Cooperative Micro-agents in Section 3.2.1
> > >
> > > By "cooperation," we refer to the indirect dependency of the Task Suggestor micro-agent on the quality of the summary provided by the Dataset Summarizer. They inherently cooperate with each other in this manner. We will clarify the term "cooperative" and elaborate on the interactions and relationships between different micro-agents.
> > >
> > > ## Relocation of Section 3.5
> > >
> > > We acknowledge the structural concern and will restructure the paper to move the proposed future work to the appropriate sections.
> > >
> > > ## Specificity in Section 4
> > >
> > > In our approach, we adhere to the complex ill-defined tasks as explained by the TELeR taxonomy paper (https://arxiv.org/abs/2305.11430), which proposes prompts with varying levels of details and styles for complex tasks. As the complexity of the goal task increases, so does the length and number of possible prompts to address the task while still missing information from users because the tasks, by definition, are ill-defined. This is the scenario we mean by the “demanding scenarios”.
> > >
> > > ## Redundancy in Section 4.2
> > >
> > > We take note of the redundancy and will revise this section to ensure it adds value to the paper.
> > >
> > > ## Dialogue Flow Control in Section 5.1
> > >
> > > In our design, the decision on which global or local micro-agent will run is contingent on the current state of the dialogue. For instance, the `Data Visualization` state has only two local micro-agents available (`Data Summarizer`, `Task Suggestor`). Our global micro-agent, `State Detector`, determines the current dialogue state, inherently deciding which local micro-agents to run.
> > >
> > > ## Appendix Sections
> > >
> > > We will revise the appendix to provide new details or merge relevant sections into the main paper, ensuring a comprehensive presentation of our work.
> > >
> > > ## Typo on pg 20
> > >
> > > This will be corrected.
> > >
> > > ## Revised TELeR Taxonomy in Section C.4
> > >
> > > In our next revision, we will update this section to reflect the correct version and provide clear examples of how the taxonomy levels translate to different input prompts.
> > >
> > > ## Temperature Setting and Randomizer's Seed
> > > We will specify these details to ensure reproducibility.

---

### Review · Reviewer_suFq · 2023-10-06

**Summary Of Contributions:**

The authors propose a pipeline for using large language models (LLMs) to automate the work currently done by data scientists. That is, the aim is to have the LLM help an end user determine what machine learning task should achieve the user’s goals for the dataset, and then determine how to set up and train the model, and summarize the results and provide recommendations to the end user. To pursue this goal, the authors propose a pipeline called VIDS (Virtual Interactive Data Scientist) that uses a sequence of prompts to get an LLM to act as an interface between the end user and a ML toolkit such as scikit-learn. The authors investigate their proposed approach with some case studies.

**Audience:**

Yes

**Broader Impact Concerns:**

Given my concerns that the paper’s statements are not backed up by evidence, there is the possibility that this paper might mislead people into thinking that current LLMs can be trusted to perform data science tasks, when in fact we do not have enough evidence to determine whether this is the case.

**Claims And Evidence:**

No

**Requested Changes:**

Unfortunately, in order for me to recommend acceptance, the paper would need to be substantially rewritten and restructured, so I think it is unlikely that changes could be made that are in the scope of a revision and that would be sufficient for me to recommend acceptance. So below I will list the changes that would be necessary for me to recommend acceptance, but with the caveat that doing all of this would be quite an extensive amount of work that seems out of scope for a revision:
- RC1: Add quantitative analysis of the pipeline’s performance. The analysis should cover a sufficiently large sample size of cases that we can obtain a reasonable estimate of how successful the approach is. At a minimum, it should include data on how accurate the model’s predictions were at each stage in the pipeline, and also how accurate the whole end-to-end system is (to address the potential issue that errors might accumulate as you go down the pipeline).
- RC2: Add some experiments testing how useful VIDS is for users who know nothing about data science. (This could be done at the same time as the experiments in RC1, by having those experiments be run by non-data-scientists, or it could be done separately). This relates to W4 above.
- RC3: Revise the paper to make sure that all of the claims it makes are backed up by evidence, removing any claims about the system’s strength that do not have evidence to back them up (see W3).

**Strengths And Weaknesses:**

Strengths:
- S1: The goal is an interesting and ambitious one that aims to use the impressive power of LLMs in a way that could make data science work more efficient and widely accessible.
- S2: The authors provide a careful breakdown of the data science analysis process. This breakdown may be useful to future authors investigating ways to make data science more efficient.
- S3: The authors provide careful analysis for each part of the data science pipeline of how to make that part accessible to an LM. In particular, one aspect that I find useful is the prompting techniques for producing machine-interpretable content, such as PeTEL data, as a way to encapsulate the LLMs decisions in a way that can then be fed to an ML toolkit.
- S4: The authors clearly lay out strengths and limitations of the approach.

Weaknesses:
- W1: The paper does not have quantitative results to support its claims. This makes it very challenging to evaluate how successful the proposed approach is. This means that unfortunately I don’t think the paper makes much headway toward answering its research questions on the bottom of page 2; we can’t tell if LLMs are feasible if we can’t get some reliable estimate of how well they perform the task; we can’t assess how accurate LLMs are without some quantitative analysis, and we can’t see what the common challenges are without a large enough sample size to determine what challenges are common. I recognize that, as the authors say, the task is ill-defined, making it difficult to evaluate; but it would still be possible to attain quantitative results by, e.g., using a sample of 100 data science scenarios and then computing statistics for how well VIDS performed on various dimensions of desired behavior (e.g., “92% accuracy at identifying the correct task.”)
- W2: The paper does provide some anecdotal results, but it is difficult to assess the LLM’s performance here because most of the concrete material for these case studies is in the appendix and is formatted in a way that makes it hard to tell if the LLM performs well; we can see the statements that the LLMs make, but we can’t tell if its decisions are in fact the right way to analyze the dataset.
- W3: The paper makes many unsubstantiated statements about the strength of the proposed approach. As a non-exhaustive list of examples, it states that the approach “**proficiently** handles the nuances of user utterances”, that “VIDS **deftly** incorporates a number of distinct states”, that “it facilitates a **seamless** dialogue flow, aligning with user needs while **enriching** their interactive experience”,that “this micro-agent generates **accurate** responses by integrating inputs from other micro-agents”, and that another component is “**ensuring** no aspect of the task is missed” (the emphases in the preceding quotes are mine).  However, there is no evidence to support the claims that it is “proficient”, “deft”, “seamless”, “enriching”, or “accurate” or that it “ensures” strong performance. To make such statements, there should be concrete evidence. For instance, to state that the system is accurate, there should be an analysis of enough different use cases that you can quantitatively compute the accuracy; or to state that it enriches the user experience there should be some sort of analysis where users are asked about their experience. These statements are therefore risky because they could mislead readers into thinking that VIDS or more impressive or advanced than it is. The authors do carefully note limitations of their approach, but it is not enough to solely list limitations in a limitations section; the rest of the paper should also be written in a way that is consistent with the project’s strengths and limitations.
- W4: One of the main motivations for the approach is that it can remove some of the need for expert data scientists in some phases of a project. However, I do not believe that the paper makes a convincing case that VIDS actually accomplishes this. Specifically, in order for this to be possible, we need to be able to trust the LLM so completely that we can be confident its recommendations are correct. Without this complete trust, we would still need a data scientist to verify that the recommendations are reasonable; the dialogues (e.g., in Table 10) do ask users for verification, but if the user doesn’t know data science, they don’t have the knowledge to really tell if the recommendations are reasonable. Therefore, in order to achieve the paper’s goal, it seems important to establish whether the LLMs perform well enough to be completely trusted, which seems unlikely given issues of hallucination in LLMs. Since the paper does not provide analysis of how reliable LLMs are, I remain unconvinced that LLMs can be a useful contribution in this area. An alternative framing could be that LLMs could be used by data scientists to speed up their work, but even this would still require some analysis to establish whether data scientists find LLMs useful for this purpose.

In sum, while I applaud the authors' ambition and their aim to explore the true extent of the usefulness of LLMs, I am concerned that the execution does not achieve the paper's stated goals.

---

> ### Author Response · Authors · 2023-10-12
> **Quantitative results are ready, User-study is ongoing**
>
> Thank you for your comprehensive review and invaluable feedback on our manuscript. We genuinely appreciate the time and effort you've invested in guiding us to enhance the quality of our work.
>
> - RC1: We wholeheartedly agree on the significance of a robust quantitative analysis. In fact, we have already performed an exhaustive quantitative analysis. Due to space constraints, we had to make the difficult decision of splitting this work into two papers, one paper focusing on the architecture and the other paper focusing on the quantitative analysis. The quantitative analysis paper is also under review at TMLR and can be found at this anonymized link: https://openreview.net/forum?id=ECVG0N8-Hs
>
>   To further clarify, the primary objective of this paper is to delineate the architecture of an actual end-to-end conversational data science pipeline. In contrast, the paper we've separately submitted to TMLR centers on a quantitative study of the VIDS agent's prediction accuracies. Given the large amounts of results we wanted to share with the community, we decided to present these facets in two distinct submissions for clarity. However, if integrating the quantitative findings from the aforementioned paper would enrich the current manuscript, we are more than willing to do so. Please advise us, and we will merge the two submissions (although it will be a little challenging).
>
> - RC2: Your emphasis on VIDS's usability, especially for those unfamiliar with data science, resonates with us. We're excited to share that we are in the midst of a user study involving a diverse group, encompassing both data science experts and novices. Given that this study is nearing its conclusion, we anticipate incorporating its insights into our manuscript within the next month. Space will be an issue again for adding more studies, please advise us regarding that.
>
> - RC3: With the enriched data and insights from RC1 and RC2, we pledge to be meticulous in our claims. Our revised manuscript will strictly adhere to claims substantiated by our quantitative and qualitative analyses, ensuring that unsupported assertions are either removed or appropriately revised.
>
> We recognize the magnitude of the revisions ahead, but given that the quantitative results are ready, we think a revision is quite feasible. We are committed to elevating our paper's quality and rigor. We eagerly anticipate your feedback on our revised manuscript and are grateful for your patience and guidance throughout this process.

---

> > ### Comment · Reviewer_suFq · 2023-10-28
> > **These directions sound promising; I can't make decisions without seeing a revised version**
> >
> > Thank you for the response! It's very helpful to know about this additional paper, and I can completely sympathize with how hard it is to write up a large-scale project; it's unfortunate that so few ML venues that can handle papers longer than 10 pages.
> >
> > Regarding RC1: It seems like there are two goals across these two papers: (i) introduce a new architecture and (ii) argue that the architecture is effective by presenting quantitative results. I think that these two things really should be done together in the same paper, because neither one is very effective without the other. If you have (i) but not (ii), it's hard for readers to appreciate why they should care about the new architecture. If you have (ii) but not (i), then it's hard for readers to appreciate the quantitative results because they won't know exactly what architecture was used to produce those results.
> >
> > So based on that, I think the most natural course would indeed be to merge the two papers into one. That would certainly run into space constraints, but there are several ways you could approach this:
> > - Move lots of the details into an appendix.
> > - Release a detailed paper as a technical report on arXiv, and then submit a condensed version to TMLR, pointing readers to the technical report in case they want more details.
> > - Note that TMLR allows submissions to be any length (https://jmlr.csail.mit.edu/tmlr/author-guide.html), so in principle you could submit the paper as, say, a 20-page paper.
> > - You could also consider submitting instead to JMLR, where longer papers are more standard.
> >
> > Alternatively, if the two papers truly do have distinct intellectual contributions, then it would make sense to keep them as two separate papers. But for that to work, they would need to be at least reasonably standalone. For instance, "Introducing “Forecast Utterance” for Conversational Data Science" definitely needs to include a description of the pipeline (maybe it already does, I have not had a chance to read it), and "Large Language Models as your Personal Data Scientist" definitely needs some quantitative results to back it up. In principle it could be fine if the quantitative results in "Large Language Models as your Personal Data Scientist" were just results from other papers that you cite; but if all the quantitative results are cited from "Introducing “Forecast Utterance” for Conversational Data Science", that would make it hard to see what additional contribution is made by "Large Language Models as your Personal Data Scientist" (this is because "Introducing “Forecast Utterance” ..." should also include a description of the pipeline, meaning that both papers would have the same basic content - description of the same pipeline and description of the same empirical results - just with different emphases, and I don't think that different emphasis is enough to merit having two papers).
> >
> > Your responses to RC2 and RC3 also sound good to me. Regarding length for RC2, note that TMLR does not have a strict length limit, so if it's necessary to go over the length limit to add a user study, that seems justified to me - just aim to have it be as concise as possible.
> >
> > Finally, as a general point, the fact that these changes would be extensive means that I would need to see an actual revision in order to be able to recommend acceptance, since (e.g.) I would need to make sure that the quantitative results convincingly supports the claims that are made. So, I think I will not be able to recommend acceptance on this current version but would be open to reviewing a revised version; if the revised version incorporates quantitative results, incorporates the user study, and revises the text in accordance with RC3, then I would be likely to recommend acceptance (but can't guarantee it without seeing the revised version).

---

### Review · Reviewer_MYpM · 2023-10-17

**Summary Of Contributions:**

The paper presents a case study in developing an automatic conversational data science assistant through large language models. It presents a conversational AI agent called VIDS that helps users conduct data science tasks, describing the collection of micro-agents that constitute the full system. The authors claim with empirical evidence that VIDS powered by LLMs show strong potential for enabling conversational data science and thus making data science explorations more accessible to the general public. It shares details of the prompts used throughout the system and a single case study with the student performance dataset.

**Audience:**

Yes

**Broader Impact Concerns:**

The authors should expand on the limitations, especially for the ethical considerations. What are the broader ethical implications of an automatic conversational data sciences system? What are the risks that we should be especially careful of in this domain? How could LLMs malfunctioning in a certain dialogue state lead to negative outcomes, such as an incorrect result summary and thus an erroneous conclusion delivered to a naive data scientist? The current write-up in this section is extremely shallow and it suggests that the authors did not consider these risks.

**Claims And Evidence:**

No

**Requested Changes:**

- RC1: In general, better motivate each design decisions. Currently, the paper reads as "we did this and this was the result", rather than a scientific exploration that goes "we tried XYZ and the best results were with X and our analysis shows that this is so because of ABC".
- RC2: Propose an evaluation framework, even if it involves human evaluation, for each micro-agent or state so that a concrete baseline is established for future work to build on. A lot of space can be made for this by removing redundancies and fancy language.
- RC3: Citation format should be (X et al., 2023) instead of X et al. (2023) when citation is not used as the subject/object.
- RC4: Contextualize the student performance dataset so that readers can understand why it is a good dataset for studying conversational data science. Even after reading the appendix, it is unclear why this is the chosen dataset when it seems like there can be many other alternatives. Unless there is a strong reason why this dataset is the most suitable, the author should consider replicating the study with at least one other dataset.
- RC5: Briefly explain the TeLER taxonomy so the paper can be more self-contained.

**Strengths And Weaknesses:**

Strengths
- S1: Describes in interesting architecture for enabling conversational data science, a novel field of study.
- S2: Details for the prompts used are shared and a case study with these prompts are shared.

Weaknesses
- W1: This paper presents a concept, rather than a scientific study. Although it is understandable that it is difficult to quantify the performance of each micro-agent, the contributions of this paper would be much more impactful if it proposed an evaluation framework for this task and established a baseline with the presented system. Without some measure of statistical reliability for each micro-agent, it is impossible for the readers to gauge whether the claims of this paper that VIDS is a promising system is supported.
- W2: There are no ablation studies that motivate the design decisions. For example, how important is it to have a dialogue summarizer? Can we provide just the last few turns of the dialogue instead?
- W3: Ethical considerations are only shallowly discussed despite the potential for severe negative outcomes if the conversational data science assistant provides an erroneous conclusion from the data.

---

> ### Author Response · Authors · 2023-10-28
> **Discussion and clarification of reviewer concerns**
>
> Thank you for your detailed feedback on our manuscript. We appreciate the time and effort you've invested in reviewing our work, and we're committed to addressing each of your concerns.
>
> ## Ablation Study and Design Decisions:
> We acknowledge the importance of understanding the contribution of each component in our system. To this end, we will include an ablation study in our revised manuscript to report the significance of each global micro-agent. This will provide a clearer picture of the necessity and impact of each component in our VIDS architecture.
>
> ## Evaluation Framework and Quantitative Analysis:
> We understand the importance of establishing a concrete baseline for our work. We have already initiated a comprehensive user study involving a diverse group of participants, including both data science experts and novices. This study is nearing its conclusion, and we anticipate incorporating its insights into our manuscript within the next month.
>
> For accuracy, we have performed an exhaustive quantitative analysis, which due to space constraints, has been presented in a separate paper currently under review at TMLR. This paper, available at [this anonymized link](https://openreview.net/forum?id=ECVG0N8-Hs), focuses on the quantitative study of the VIDS agent's prediction accuracies. If integrating the quantitative findings from this paper would enrich the current manuscript, we are open to merging the two submissions, though it will be challenging due to space constraints.
>
> ## Citation Format:
> We apologize for the oversight and will correct the citation format as per your recommendation.
>
> ## Dataset Contextualization:
> The quantitative analysis paper at [this anonymized link](https://openreview.net/forum?id=ECVG0N8-Hs) includes VIDS’s performance on three different datasets: Online Delivery, Student Performance, and Flight Delay datasets, demonstrating VIDS's adaptability across multiple datasets. The foundational idea behind our work is to showcase VIDS's capability to adapt to any dataset and collaboratively formulate an ML task with the user. The dataset presented in the experiment was chosen randomly to exemplify this adaptability. We will clarify this in our revised manuscript and ensure that our rationale behind dataset selection is communicated more transparently.
>
> ## TeLER Taxonomy:
> To make our paper more self-contained, we will include a description of the TELeR taxonomy, ensuring readers have all the necessary context to understand our work without external references.
>
> ## Ethical Considerations:
> We recognize the importance of discussing the ethical implications of our work, especially given the potential consequences of erroneous conclusions. We will expand on this section in our revised manuscript, addressing the potential risks and our mitigation strategies.
>
> In conclusion, we are committed to making the necessary revisions to address your concerns and enhance the clarity and depth of our manuscript. We appreciate your guidance throughout this review process.

---

### Decision · Action_Editor_bs1z · 2023-12-07

**Recommendation:** Reject

**Comment:**

All the reviewers commended the authors for working on a potentially impactful effort of leveraging LLMs to facilitate data science tasks. The authors also made a nice connection to AutoML. Reviewer suFq also noted that "The authors provide a careful breakdown of the data science analysis process. This breakdown may be useful to future authors investigating ways to make data science more efficient." Reviewer tDKy mentioned that "References quite relevant literature for a detailed systems-level approach to solving a data science problem." Reviewer MYpM concurred that "While the concept is interesting, a lack of quantitative results make the claims of the effectiveness of the proposed system unconvincing."

However, all the reviewers are concerned with the lack of quantitative results. In the recommendation, Reviewer suFq stated that "The lack of quantitative results makes me unable to recommend acceptance." Reviewer tDKy is concerned with the "rigor for metrics and evaluation of experimental results". From the discussions between the authors and the reviewers, it seems that the authors have a concurrent paper under review at TMLR contains more quantitative evaluation of the work. All the reviewers suggest that the authors should submit a single expanded version of the work instead of having two separate submissions that each can't stand on their own.

**Audience:**

The work is potentially interesting, as all the reviewers agreed. However, without quantitative evaluation, it is hard for readers to take these results with confidence.

**Claims And Evidence:**

All the reviewers pointed out that the major issue with the paper is the lack of quantitative empirical results to support their claims. The authors brought up that they have a concurrent paper under review at TMLR that is more focused on the empirical results of the work. It seems that the census of the reviewers is that the two submissions should be merged as it is hard for them standalone.

**Resubmission Of Major Revision:**

The authors may consider submitting a major revision at a later time.